# Assessment of the effect of therapy in a rat model of glioblastoma using [18F]FDG and [18F]FCho PET compared to contrast-enhanced MRI

Julie Bolcaen[1]*, Benedicte Descamps[2], Karel Deblaere[3], Filip De Vos[4], Tom Boterberg[5], Giorgio Hallaert[6], Caroline Van den Broecke[7], Christian Vanhove[2], Ingeborg Goethals[8]

1 Radiation Biophysics Division, Department of Nuclear Medicine, National Research Foundation iThemba LABS, Faure, South Africa, 2 Department of Electronics and Information Systems, IBiTech-MEDISIP, Ghent University, Ghent, Belgium, 3 Department of Radiology, Ghent University Hospital, Ghent, Belgium, 4 Department of Radiopharmacy, Ghent University, Ghent, Belgium, 5 Department of Radiation Oncology, Ghent University Hospital, Ghent, Belgium, 6 Department of Neurosurgery, Ghent University Hospital, Ghent, Belgium, 7 Department of Pathology, Ghent University Hospital, Ghent, Belgium, 8 Department of Nuclear Medicine, Ghent University Hospital, Ghent, Belgium

* jbolcaen@tlabs.ac.za

**Data Availability Statement:** All relevant data are within the paper and its Supporting Information files.

## Abstract

### Objective

We investigated the potential of [18F]fluorodeoxyglucose ([18F]FDG) and [18F]Fluoromethyl-choline ([18F]FCho) PET, compared to contrast-enhanced MRI, for the early detection of treatment response in F98 glioblastoma (GB) rats.

### Methods

When GB was confirmed on T2- and contrast-enhanced T1-weighted MRI, animals were randomized into a treatment group (n = 5) receiving MRI-guided 3D conformal arc micro-irradiation (20 Gy) with concomitant temozolomide, and a sham group (n = 5). Effect of treatment was evaluated by MRI and [18F]FDG PET on day 2, 5, 9 and 12 post-treatment and [18F]FCho PET on day 1, 6, 8 and 13 post-treatment. The metabolic tumor volume (MTV) was calculated using a semi-automatic thresholding method and the average tracer uptake within the MTV was converted to a standard uptake value (SUV).

### Results

To detect treatment response, we found that for [18F]FDG PET (SUV$_{mean}$ x MTV) is superior to MTV only. Using (SUV$_{mean}$ x MTV), [18F]FDG PET detects treatment effect starting as soon as day 5 post-therapy, comparable to contrast-enhanced MRI. Importantly, [18F]FDG PET at delayed time intervals (240 min p.i.) was able to detect the treatment effect earlier, starting at day 2 post-irradiation. No significant differences were found at any time point for both the MTV and (SUV$_{mean}$ x MTV) of [18F]FCho PET.

**Funding:** Lux Luka Foundation supported this study financially. Funds were received by Prof. I. Goethals and Prof. T. Boterberg. The sponsor did not play a role in study design and analysis.

**Competing interests:** The authors have declared that no competing interests exist.

## Conclusions

Both MRI and particularly delayed [18F]FDG PET were able to detect early treatment responses in GB rats, whereas, in this study this was not possible using [18F]FCho PET. Further comparative studies should corroborate these results and should also include (different) amino acid PET tracers.

## Introduction

In the US, 84,170 new cases of primary brain and other central nervous system tumors are estimated to be diagnosed in 2021. Glioblastoma (GB) has the highest number of cases of all malignant tumors, with 12,970 cases projected in 2021 [1]. Surgical resection remains the most effective treatment for gliomas. It has been shown that patients who had a gross total resection have a better response to subsequent adjuvant treatments than those who underwent a partial resection or biopsy only [2]. However, in about half of the newly diagnosed patients, gross total resection is not possible [3]. In 2005, Stupp *et al.* established the superiority of surgery and combined chemoradiation therapy with temozolomide (TMZ) over surgery and radiation therapy (RT) alone. As a result, for newly diagnosed glioblastoma (GB) patients with a good performance status, the standard of care now includes maximal surgical resection followed by combined external beam RT (60 Gy in 30 fractions) and TMZ [4–6]. Despite the combined treatment, the clinical course of GB is usually rapid and provides only moderate survival benefit with a median survival of approximately 1 year [5]. Hence, the assessment of early treatment response is crucial allowing early change of management and the discontinuation of ineffective treatment and treatment related adverse effects/events. Moreover, the identification of early treatment failure may reduce costs because new systemic treatments (e.g. bevacizumab) are considerably more expensive than conventional alkylating chemotherapy (e.g. lomustine) [7–9].

In the clinic, MRI is the imaging modality of choice for therapy response assessment in glioma patients. New response criteria for Response Assessment in Neuro-Oncology (RANO) were introduced in 2010, including the tumor size (in 2D) as measured on T2- and Fluid Attenuated Inversion Recovery (FLAIR)-weighted images, in addition to the contrast-enhancing tumor part [10]. However, tumor cells have been found at quite a distance from the contrast enhancing region on MRI [11]. Furthermore, 20 to 30% of patients undergoing a first post-irradiation MRI show increased contrast enhancement that eventually subsides without any change in therapy. This phenomenon is well-known as so-called pseudoprogression. On the other hand, pseudoresponse also occurs when antiangiogenic treatment is given, especially when targeting vascular endothelial growth factor (VEGF) with for example bevacizumab, and the VEGF receptor, with cediranib. Antiangiogenic treatment usually causes a marked decrease in contrast enhancement as early as 1 to 2 days after the initiation of the treatment. These apparent radiologic responses may be partly the result of normalization of abnormally permeable tumor vessels and may therefore not indicate a true/genuine antitumor effect *per se*. Hence, pseudoprogression and pseudoresponse complicate treatment response assessment in glioma patients using conventional MRI [10, 12]. In addition, therapy-related effects on normal brain tissue, such as radiation necrosis, inflammation and postsurgical changes can also result in increased enhancement or FLAIR/T2 hyperintense signal abnormalities. This also adds to the complexity of assessing treatment response and tumor recurrence that is particularly important in patients with high-grade gliomas for whom the treatment of tumor

recurrence is always urgent [10, 13–15]. Similarly, pseudoprogression is also common when immunotherapy, which is being increasingly applied in neuro-oncology, is administered [8, 16–18].

To address the above-mentioned drawbacks, incorporating techniques measuring treatment induced changes in tumor biology may help [3, 19, 20]. Importantly, PET enables visualization of biological changes preceding anatomical changes. And with an overall increasing access to PET tracers, three indications for PET imaging are of particular interest: (i) the identification of tumor tissue, including the delineation of tumor volume, (ii) the differentiation of treatment-related changes from tumor progression at follow-up, and (iii) the assessment of treatment response for predicting outcome [21]. For these purposes, multiple PET tracers have been proposed, such as [18F]fluorodeoxyglucose ([18F]FDG), [18F]Fluoroethyltyrosine ([18F] FET), [18F]fluoroazomycin arabinoside ([18F]FAZA), 3,4-dihydroxy-6-[18F]-fluoro-l-phenylalanine ([18F]FDOPA) and [18F]Fluoromethylcholine ([18F]FCho) [21–24]. Currently only two of these are frequently used in the clinic, namely [18F]FDG and [18F]FET. [18F]FDG PET measures cellular glucose metabolism as a function of the hexokinase enzyme. However, due to its high uptake in normal brain parenchyma, the localization and the delineation of brain tumors is often difficult [23, 24]. It was shown that delineation of gliomas was improved by extending the interval between the administration of [18F]FDG and PET acquisition, the so called "dual phase imaging" [25–27]. Also, increased 18F-FDG uptake in inflammatory tissue hampers its specificity. Therefore, new PET tracers beyond [18F]FDG such as radiolabeled amino acids were developed showing an increased contrast between brain tumors and normal brain tissue. The diagnostic potential of [18F]FET PET in brain tumors is well documented and the RANO working group has recommended amino acid PET as an additional tool in the diagnostic assessment of brain tumors [28]. Also, a superior delineation of gliomas by [18F] FET PET compared with MRI and a promising role for the distinction between tumor recurrence and aspecific post-therapeutic changes has been shown [7, 29, 30]. Hypoxia imaging, using [18F]FAZA as a PET tracer may also have clinical relevance because tumor aggressiveness, failure to achieve local tumor control and an increased rate of tumor recurrence are all associated with hypoxia [22, 31, 32]. A downside of [18F]FAZA PET is that optimal imaging is performed a few hours post-injection and that the degree of hypoxia can theoretically fluctuate, influenced by therapy and the presence of acute versus chronic hypoxia [33]. Finally, positron-labeled choline analogues appear to be successful as oncological PET probes because a major hallmark of cancer cells is increased lipogenesis, resulting in a high tumor-to-normal brain tissue contrast [34, 35]. Recently, the current status of choline PET in neuro-oncology was reviewed. The major advantage of choline tracers is the very low uptake in normal white and grey matter and its accessibility because of its use in the management of castration resistant prostate cancer. The metabolic information acquired by [18F]FCho PET has been shown to be able to distinguish high-grade glioma, brain metastases and benign lesions and to identify the most malignant areas for stereotactic sampling [35–38]. Grech-Sollars *et al.* concluded that [18F]FCho PET was able to differentiate WHO (World Health Organization) grade IV from grade II and III tumors, whereas MR spectroscopy differentiated grade III/IV from grade II tumors [39]. Recently, the potential use of [18F]FCho PET/CT in the intraoperative management or radio-surgical approaches for glioma has been suggested, including intraoperative guidance in conjunction with MR spectroscopy [38, 40, 41].

Previously, our group used the orthotopic allograft F98 GB rat model to mimic GB treatment in patients. The GB F98 rat model exhibited features of human GB with regard to its aggressiveness, histological appearance and lack of immunogenicity [42]. To enable more precise irradiation of the target volume in small animals, precision image-guided small animal radiation research platforms were developed, such as the Small Animal Radiation Research

Platform (SARRP, Xstrahl[R], Surrey, UK). Using the F98 GB rat model and the SARRP, we described and validated magnetic resonance imaging (MRI)-guided 3D conformal arc RT with concomitant chemotherapy to bridge the gap between radiation technology in the clinic and preclinical techniques [43]. In this study, this methodology was applied to investigate the potential of [18F]FDG and [18F]FCho PET, compared to contrast-enhanced MRI, to detect the early effect of combined radiation and TMZ treatment in the F98 GB rat model. In addition, we also investigated which modality is best suited for the early detection of treatment response.

## Materials and methods

The study was approved by the Ghent University Ethical Committee for animal experiments (ECD 09/23-A). All animals were kept under environmentally controlled conditions (12-h normal light/dark cycles, 20˚C– 24˚C, and 40–70% relative humidity) with food and water ad libitum. Follow-up of all animals was done by monitoring their body weight, food, water intake and their activity and normal behavior. The method of euthanasia was a lethal dose of pentobarbital sodium (180 mg/kg). Euthanasia was performed prior to the experimental end-point if a decline of 20% body weight was observed or when the normal behavior severely deteriorated (e.g. lack of grooming).

### F98 GB rat model

F98 Glioma cells (ATCC, 20 000 in 5 μl cell suspension) were inoculated 2 mm posterior and 2.5 mm lateral to the bregma in the right frontal hemisphere of female Fischer F344 rats (Charles River[R]) (n = 10, body weight 173 ± 11 g, mean ± SD). Full details of the protocol can be found in our previous publications [43–45]. For inoculation, rats were anesthetized with ketamine/xylazine (4/3; 0.13 ml/100 g). Post-surgery, a close follow-up of the animals was performed (body temperature, wound healing and behavior). Animals were kept separately post-inoculation.

### MRI-guided 3D conformal arc micro-irradiation

Nine days post-inoculation, MRI was performed using a 7 tesla preclinical MRI system (PharmaScan 70/16, Bruker BioSpin, Ettlingen, Germany). The rats (fixed on the multimodality bed) were anesthetized with isoflurane mixed with oxygen at a flow rate of 0.3 L/min (induction 5%, maintenance 1.5%) and covered with a heated blanket. The bed was placed in a holder with fixed rat brain surface coil (Rapid Biomedical, Rimpar, Germany) that was positioned in a 72 mm rat whole body transmitter coil (Rapid Biomedical, Rimpar, Germany). A localizer scan was performed followed by a T2-weighted spin-echo scan (TR/TE 3661/37.1 ms, 109 μm isotropic in-plane resolution, 4 averages, TA 9'45") to assess tumor growth. Secondly, gadolinium-containing contrast (Dotarem, Guerbet, France; 0,4 mL/kg) was injected intravenously. Fifteen minutes later a contrast-enhanced T1-weighted spin echo sequence (TR/TE 1539/9.7 ms, 117 μm isotropic in-plane resolution, 3 averages, TA 4'15") was performed. Typical T2- and contrast-enhanced T1-weighted MR images are shown in Fig 1A and 1B. For the treatment group (n = 5), when tumor growth was confirmed on MRI, the animal was transported to the table of the SARRP. A high-resolution treatment planning CT scan was performed, using an aluminum filter of 1 mm and a 20 x 20 cm (1024 x 1024 pixel) amorphous Si flat panel detector. A total of 720 projections were acquired over 360˚ and reconstructed with an isotropic voxel size of 0.2 mm, see Fig 1C. The tube voltage and tube current were set at 80 kV and 0.6 mA, respectively. The CT and the T1-weighted contrast-enhanced MRI scan were imported into 3D slicer v3.6.31 (www.slicer.org) and co-registered manually with rigid body

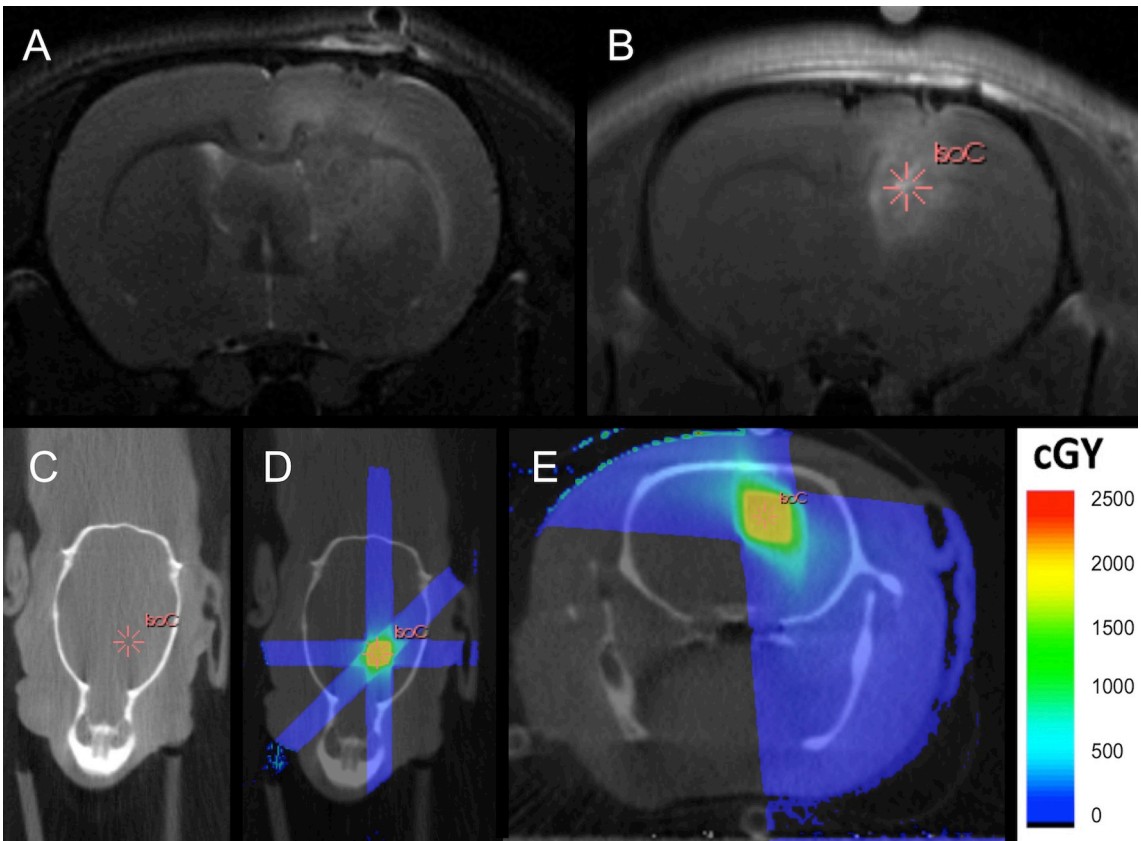

**Fig 1. MRI-guided 3D conformal arc micro-irradiation.** T2-weighted (a) and contrast-enhanced T1-weighted MRI (b) clearly visualizing a rat F98 brain tumor. No tumor is visible on the high-resolution treatment planning CT scan (c). The isocenter for irradiation is selected in the center of the contrast enhancing tumor part (b-e). Using a 3 x 3 mm collimator, a dose of 20 Gy is delivered applying 3 non-coplanar arcs (e).

transformations. By overlaying the increased signal intensity of the skull on CT with black signal on MRI and using multimodality markers, a precise fusion could be achieved. Using the treatment planning software (Muriplan, Xstrahl®, Surrey, UK), dose plans were calculated to deliver 20 Gy to the target tumor volume using a 3 x 3 mm collimator applying three arcs, one covering an angle of 90˚ with the couch at 0˚ and two covering angles of 60˚ with the couch at 45˚ and 90˚, see Fig 1D and 1E. Concomitant chemotherapy was administered using intraperitoneal injections of 29 mg/kg TMZ (Sigma-Aldrich®) dissolved in saline with 25% dimethyl-sulfoxide (DMSO, Sigma-Aldrich®) during 5 days starting at the day of irradiation [46, 47]. In the control group (n = 5), the animals received intraperitoneal injections of an equal amount of DMSO and saline on 5 consecutive days. Irradiation was not performed in the control group.

## Evaluation of tumor growth using MRI

Follow-up MRI scans were acquired 2, 5, 9 and 12 days after the start of treatment, see Fig 2. Two animals in the treatment group had an additional scan on day 15 and day 21 because of a stable follow-up. T2- and contrast-enhanced T1-weighted spin echo sequences were recorded. Using the PMOD software (version 3.31, PMOD technologies®, Zürich, Switzerland), the volume of the tumor was determined by manually drawing volumes of interest around the tumor on the contrast-enhanced T1-weighted MR images.

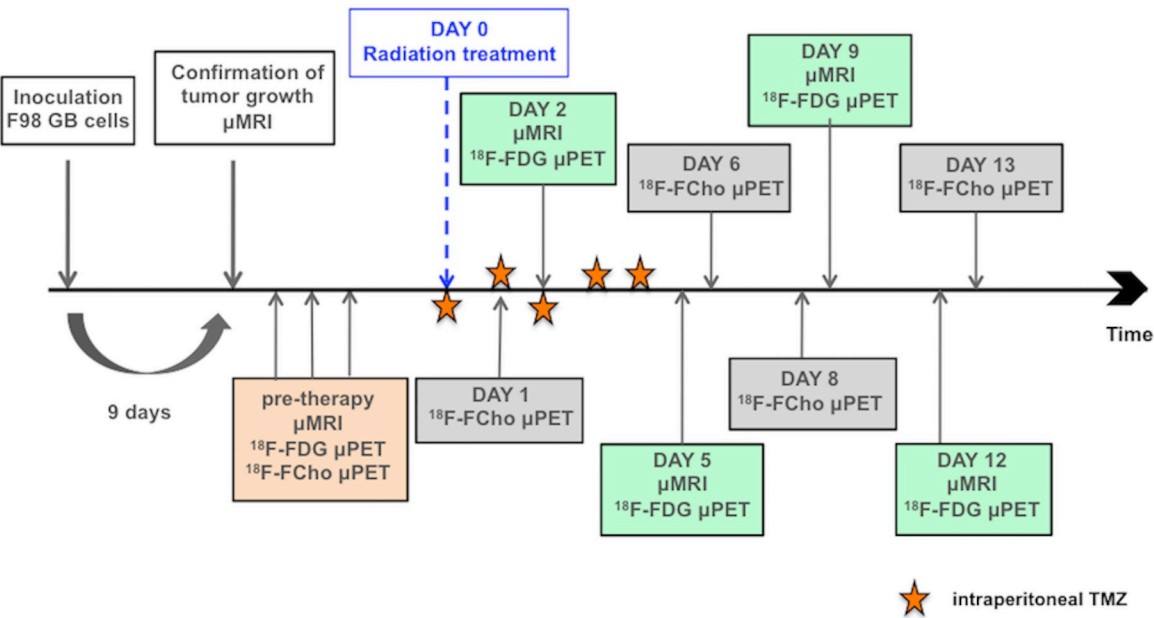

**Fig 2. Study scheme for the assessment of treatment effects using MRI, [$^{18}$F]FDG PET and [$^{18}$F]FCho PET.**

## Assessment of biological tumor response using [$^{18}$F]FDG and [$^{18}$F]FCho PET

The assessment of the biological response was evaluated by small animal PET using [$^{18}$F]FDG and [$^{18}$F]FCho. [$^{18}$F]FDG scans were performed 2, 5, 9 and 12 days after the start of treatment, while [$^{18}$F]FCho scans were performed 1, 6, 8 and 13 days after the start of treatment. These time points were arbitrarily chosen because, empirically, GB rats survived approximately 14 days after the start of treatment [43]. Also, it is worth mentioning that logistically [$^{18}$F]FDG and [$^{18}$F]FCho PET scanning was not possible on the same day. An overview of the complete imaging scheme is shown in Fig 2 and an overview of the data is listed in Table 1.

**Table 1. Overview of the μPET data in the longitudinal PET study.**

| Therapy | PRE-THERAPY | | | DAY 1 | DAY 2 | | DAY 5 | | DAY 6 | DAY 8 | DAY 9 | | DAY 12 | | DAY 13 |
|---|---|---|---|---|---|---|---|---|---|---|---|---|---|---|---|
| 0 = Control | | Early | Late | | Early | Late | Early | Late | | | Early | Late | Early | Late | |
| 1 = RT+TMZ | $^{18}$F-FCho | $^{18}$F-FDG | $^{18}$F-FDG | $^{18}$F-FCho | $^{18}$F-FDG | $^{18}$F-FDG | $^{18}$F-FDG | $^{18}$F-FDG | $^{18}$F-FCho | $^{18}$F-FCho | $^{18}$F-FDG | $^{18}$F-FDG | $^{18}$F-FDG | $^{18}$F-FDG | $^{18}$F-FCho |
| 0 | ✓ | ✓ | ✓ | ✓ | ✓ | ✓ | ✓ | ✓ | ✓ | ✓ | ✓ | ✓ | ✓ | ✓ | - |
| 0 | ✓ | ✓ | ✓ | ✓ | ✓ | ✓ | ✓ | ✓ | ✓ | ✓ | ✓ | ✓ | ✓ | ✓ | - |
| 0 | ✓ | ✓ | ✓ | ✓ | ✓ | ✓ | ✓ | - | - | - | - | - | ✓ | ✓ | ✓ |
| 0 | ✓ | ✓ | ✓ | ✓ | ✓ | ✓ | ✓ | ✓ | ✓ | - | ✓ | ✓ | - | - | - |
| 0 | ✓ | ✓ | ✓ | ✓ | ✓ | ✓ | ✓ | - | ✓ | ✓ | ✓ | ✓ | ✓ | ✓ | ✓ |
| 1 | ✓ | ✓ | - | ✓ | ✓ | ✓ | ✓ | ✓ | ✓ | ✓ | ✓ | ✓ | ✓ | ✓ | - |
| 1 | ✓ | ✓ | ✓ | ✓ | ✓ | ✓ | ✓ | ✓ | ✓ | ✓ | ✓ | ✓ | ✓ | ✓ | ✓ |
| 1 | ✓ | ✓ | ✓ | ✓ | ✓ | ✓ | ✓ | ✓ | ✓ | ✓ | ✓ | ✓ | ✓ | ✓ | ✓ |
| 1 | ✓ | ✓ | ✓ | - | ✓ | ✓ | ✓ | ✓ | ✓ | - | ✓ | ✓ | ✓ | ✓ | - |
| 1 | ✓ | ✓ | ✓ | ✓ | ✓ | ✓ | ✓ | - | ✓ | ✓ | ✓ | ✓ | ✓ | ✓ | ✓ |

Radiation therapy (RT), Temozolomide (TMZ), $^{18}$F-fluorodeoxyglucose ($^{18}$F-FDG), $^{18}$F-Fluoromethylcholine ($^{18}$F-FCho)

Dynamic PET images were acquired in list mode using a dedicated small animal PET scanner (FLEX Triumph II, TriFoil Imaging®, Northridge, CA, USA). Animals were anesthetized with 2% isoflurane mixed with oxygen (0.3 L/min). A 30-Gauge needle connected to a 10 cm long tube was inserted into the tail vein, enabling the injection of the radioactive tracer ($37.89 \pm 0.35$ MBq [18F]FDG and $39.55 \pm 0.37$ [18F]FCho (mean $\pm$ SE) dissolved in 200 µL saline). The total acquisition time was 20 min for [18F]FCho PET due the fast kinetics of [18F]FCho and 60 min for conventional [18F]FDG PET. In addition, a 30-min [18F]FDG PET scan was acquired 240 min after [18F]FDG administration (delayed imaging). All PET scans were reconstructed into a $200 \times 200 \times 64$ matrix by a 2D Maximum Likelihood Expectation Maximization (MLEM) algorithm (LabPET Version 1.12.1, TriFoil Imaging®, Northridge, CA, USA) using 60 iterations and a voxel size of $0.5 \times 0.5 \times 1.157$ mm. Identical reconstruction parameters were applied for [18F]FDG and [18F]FCho PET. The dynamically acquired PET data were reconstructed into $6 \times 20$ s, $3 \times 1$ min, $3 \times 5$ min, $2 \times 20$ min time frames for [18F]FDG scans and $6 \times 20$ s, $3 \times 1$ min, $1 \times 5$ min, $1 \times 10$ min time frames for [18F]FCho scans.

The metabolic tumor volume (MTV) was calculated based on a semi-automatic thresholding method using the PMOD software (version 3.405, PMOD technologies®, Zürich, Switzerland). MTV was defined on the last time frame of the dynamic [18F]FDG PET (40–60 min post-injection), the delayed [18F]FDG PET (240 min post-injection) and on the last time frame of the dynamic [18F]FCho scan (10–20 min post-injection). First, a circular VOI is manually placed over a region with an increased tracer uptake excluding non-specific uptake, such as uptake in the scalp. Within this VOI, MTV was defined as all voxels with an uptake $\geq 60\%$ and $\geq 50\%$ of the maximum uptake for [18F]FDG and [18F]FCho, respectively. The selection of the thresholds was done arbitrarily and based on visual inspection of the [18F]FDG PET scan 40–60 min post-injection, the delayed [18F]FDG PET scan 240–270 min post-injection and the [18F]FCho PET scan 10–20 min post-injection, see Fig 3. Average tracer uptake within the MTV was converted to a standard uptake value (SUV) according to the following formula:

$$SUV = \left( \frac{\text{uptake in the MTV } \left(\frac{Bq}{ml}\right)}{\text{injected activity } (Bq)} \right) x \text{ body weight } (g)$$

Injected activity was corrected for decay and residual activity in the syringe. In addition to the MTV, the SUV$_{max}$, tumor-to-normal brain ratio (TBR$_{max}$) and (MTV x SUV$_{mean}$) were

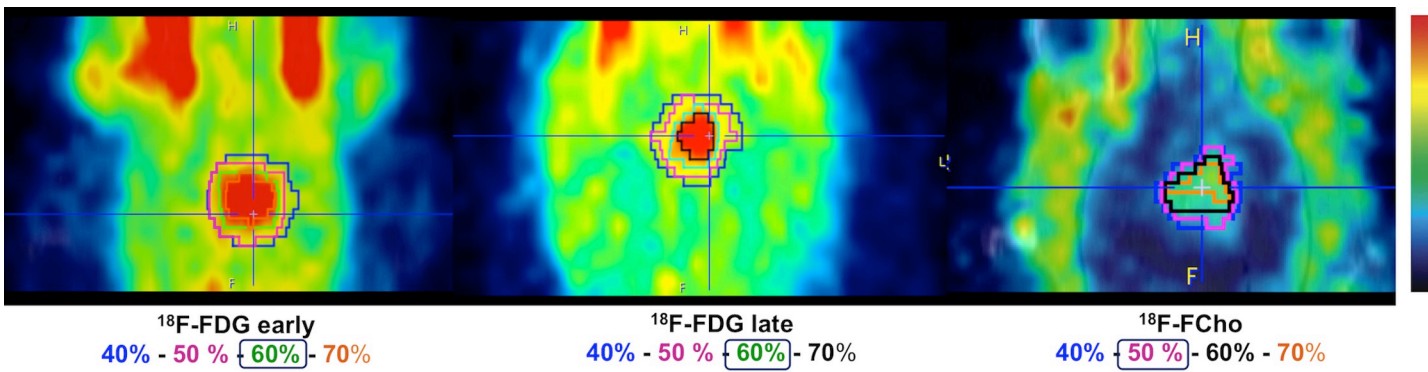

**Fig 3. Selection of the threshold for defining the metabolic tumor volume (MTV) on conventional [18F]FDG PET (left), delayed [18F]FDG PET (center) and [18F]FCho PET (right).** Different thresholds ($\geq$40-50-60-70%) are contoured. For both [18F]FDG PET (42 MBq, 40–60 min post-injection and 240 min post-injection), the threshold contouring $\geq 60\%$ of the maximum uptake deemed most appropriate (green VOI). For [18F]FCho PET (41 MBq, 10–20 min post-injection), a threshold contouring $\geq 50\%$ of the maximum uptake deemed most appropriate (purple VOI). Inhalational anesthesia with isoflurane (2%) mixed with oxygen (0.3 L/min) was used.

calculated and included in the analysis. The $TBR_{max}$ was defined as the ratio of the $SUV_{max}$ of the tumor MTV to the $SUV_{mean}$ of the reference VOI located in the contralateral occipital normal brain region.

Clinical MRI and PET images used to compare clinical and preclinical [18F]FCho PET were data from [48]. The scans were acquired using a PET Allegro system (Philips Healthcare, Cleveland, Ohio, USA). MR examinations were performed on a 3 T Siemens Trio Tim whole-body scanner (Erlangen, Germany). For details, see [48].

## Autoradiography and Evans Blue (EB) staining

To evaluate non-specific [18F]FCho uptake due to blood–brain barrier (BBB) breakdown, we performed autoradiography and analyzed EB extravasation of a F98 GB rat tumor on day 16 after inoculation, as described in [45]. 4% EB (Sigma-Aldrich®) dissolved in saline at a concentration of 4 mL per kg of body weight was injected intravenously (t = 0 min). [18F]FCho was injected (20.35 MBq, t = 5 min). At t = 60 min, the rat was euthanized, and dissected rat brains were instantly frozen in isopentane (VWR®) cooled by liquid nitrogen for 2 min followed by 30 min incubation at -20˚C. The brains were then cut into 20 μm serial sections using a cryostat (Leica®, CM3050S), with alternating slides for fluorescent staining and hematoxylin and eosin (H&E) stain. The H&E sections were dried prior to fixation in 4% paraformaldehyde. The slices for autoradiography were placed on a Super Resolution storage phosphor screen (in red lighted room) and incubated for 2.5 h. The film was scanned using the PerkinElmer Cyclone Plus (600 dpi). A picture was taken of the frozen brain tissue (Sony®), and TRITC (tetramethylrhodamine isothiocyanate) fluorescently labeled sections were imaged with a BD pathway 435 automated imaging system (Becton Dickinson) equipped with a 10× objective. A montage of 20×15 images provided a complete overview of the brain section. Using the PMOD software, the HE and AR image were manually co-registered and the tumor volume of interest (VOI) was manually drawn on the HE image and transferred to the AR image. The normal brain VOI consists of a 5 x 5 mm square placed in the contralateral normal brain.

## Statistical analysis

Statistical analysis of the MRI and PET-derived variables (MRGd tumor volumes, MTV, $SUV_{max}$, $TBR_{max}$ and MTV x $SUV_{mean}$) between the control and treatment group were analyzed by the Mann-Whitney U non-parametric test. Statistical analysis of longitudinal differences within each group was performed using the Wilcoxon Signed Rank test and the Friedman test. A probability value of $p < 0.05$ was considered statistically significant.

# Results

## Assessment of the effect of image-guided irradiation using MRI

Because tumor volumes in individual animals were variable, tumor volume after the start of irradiation were normalized to the MRI tumor volume before starting therapy. The evolution of the normalized MRI tumor volumes is shown in Fig 4. The normalized MRI tumor volume was significantly different between control and treated group on day 5 (p = 0.008), day 9 (p = 0.016) and day 12 (p = 0.032) post-therapy (see asterisk in Fig 4 and Table 2).

## Assessment of biological response of the tumor using [18F]FDG and [18F]FCho PET

Longitudinal [18F]FDG and [18F]FCho PET scans were acquired on all the rats in the control group and 5 rats in the treatment group. An overview of the data is listed in Table 1. Missing

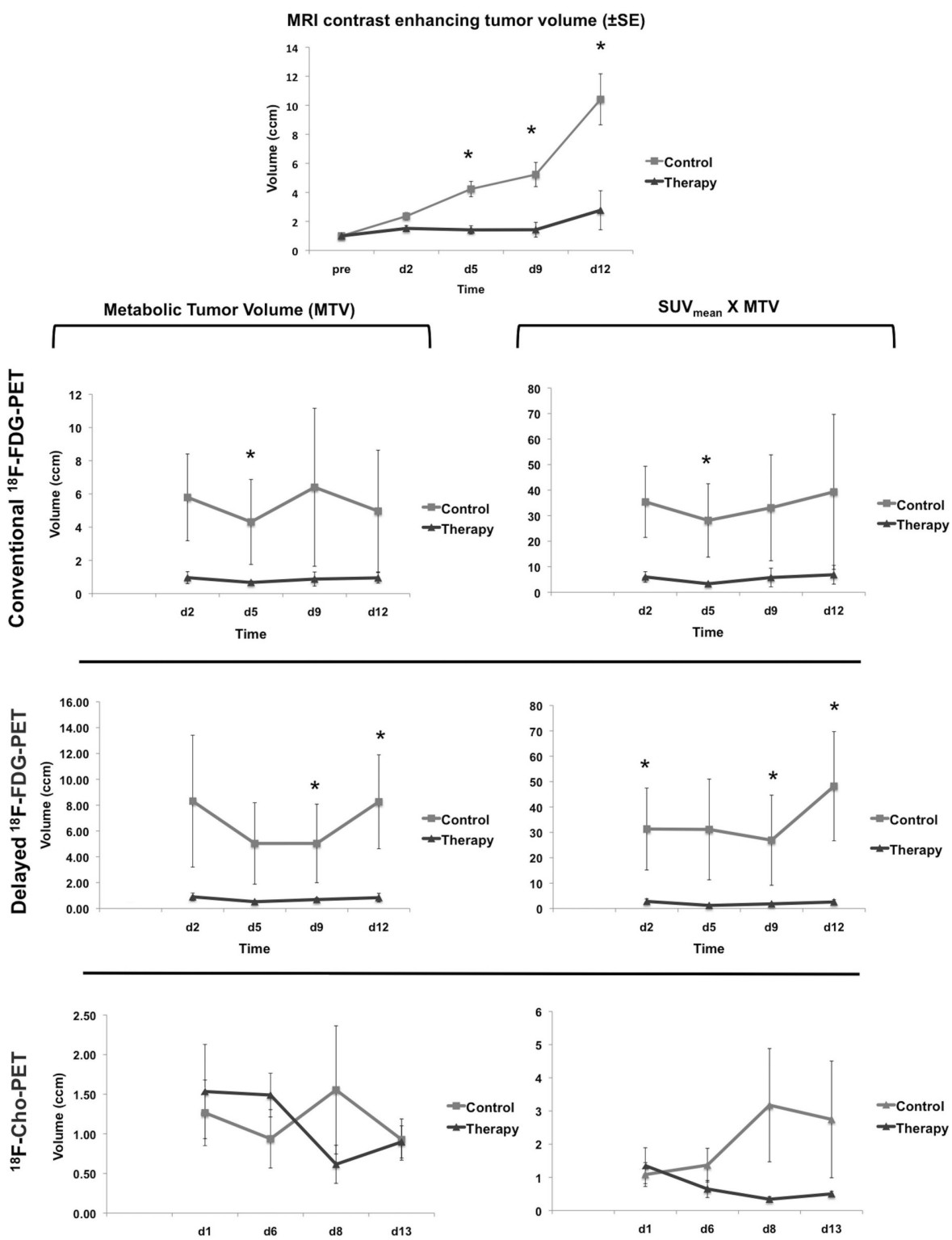

**Fig 4. Evolution of tumor growth based on MRI and PET.** Evolution of the mean tumor volumes (±SE) on T1-weighted contrast-enhanced MR images in both treatment groups. Evolution of the metabolic tumor volume (MTV) and (SUV$_{mean}$ x MTV) of conventional and delayed [$^{18}$F]FDG and [$^{18}$F]FCho PET in the control and treatment group (±SE) are given. Significant differences between the control and treatment groups ($p < 0.05$) are marked with an asterisk (*).

**Table 2. Assessment of the effect of therapy using MRI and PET.** Differences between treatment and control groups at different time points during longitudinal follow-up using the Mann-Whitney U test.

| Normalized variable | | | Time point | Mann-Whitney U exact Sig. (one-tailed) | DMSO Mean ± SE | Therapy Mean ± SE |
|---|---|---|---|---|---|---|
| MRI (Gd tumor volume)$^£$ | | | Day 2 | 0.056 | 2.36 ± 0.25 | 1.52 ± 0.20 |
| | | | Day 5 | 0.008* | 4.23 ± 0.53 | 1.41 ± 0.28 |
| | | | Day 9 | 0.016* | 5.23 ± 0.84 | 1.43 ± 0.51 |
| | | | Day 12 | 0.032* | 10.41 ± 1.76 | 2.77 ± 1.35 |
| PET (MTV) $^£$ | Conventional $^{18}$F-FDG PET | | Day 2 | 0.151 | 5.79 ± 2.61 | 0.96 ± 0.36 |
| | | | Day 5 | 0.016* | 4.31 ± 2.56 | 0.67 ± 0.10 |
| | | | Day 9 | 0.063 | 6.40 ± 4.76 | 0.87 ± 0.43 |
| | | | Day 12 | 0.190 | 4.97 ± 3.67 | 0.94 ± 0.32 |
| | Delayed $^{18}$F-FDG PET | | Day 2 | 0.151 | 8.31 ± 5.10 | 0.90 ± 0.29 |
| | | | Day 5 | 0.114 | 5.03 ± 3.15 | 0.52 ± 0.11 |
| | | | Day 9 | 0.032* | 5.04 ± 3.04 | 0.69 ± 0.14 |
| | | | Day 12 | 0.032* | 8.26 ± 3.63 | 0.84 ± 0.33 |
| | $^{18}$F-FCho PET | | Day 1 | 0.063 | 1.27 ± 0.41 | 1.53 ± 0.59 |
| | | | Day 6 | 0.111 | 0.94 ± 0.37 | 1.49 ± 0.28 |
| | | | Day 8 | 0.057 | 1.55 ± 0.81 | 0.62 ± 0.24 |
| | | | Day 13 | 0.700 | 0.93 ± 0.26 | 0.90 ± 0.20 |
| PET ($SUV_{mean}$ x MTV) | Conventional $^{18}$F-FDG PET | | Day 2 | 0.056 | 35.41 ± 13.93 | 6.00 ± 2.06 |
| | | | Day 5 | 0.008* | 28.14 ± 14.37 | 3.32 ± 0.54 |
| | | | Day 9 | 0.063 | 33.08 ± 20.75 | 5.77 ± 3.65 |
| | | | Day 12 | 0.111 | 39.33 ± 30.35 | 6.87 ± 3.67 |
| | Delayed $^{18}$F-FDG PET | | Day 2 | 0.032* | 31.33 ± 16.14 | 2.784 ± 1.03 |
| | | | Day 5 | 0.057 | 31.15 ± 19.86 | 1.20 ± 0.20 |
| | | | Day 9 | 0.032* | 26.91 ± 17.76 | 1.82 ± 0.38 |
| | | | Day 12 | 0.016* | 48.20 ± 21.51 | 2.56 ± 0.92 |
| | $^{18}$F-FCho PET | | Day 1 | 0.556 | 1.08 ± 0.36 | 1.35 ± 0.54 |
| | | | Day 6 | 0.413 | 1.37 ± 0.51 | 0.65 ± 0.26 |
| | | | Day 8 | 0.200 | 3.18 ± 1.71 | 0.34 ± 0.08 |
| | | | Day 13 | 0.200 | 2.75 ± 1.76 | 0.50 ± 0.08 |

* = p ≤ 0.05

$^£$ (normalized to the volume before starting therapy), gadolinium (Gd), metabolic tumor volume (MTV), standard uptake value (SUV), $^{18}$F-fluorodeoxyglucose ($^{18}$F-FDG), $^{18}$F-Fluoromethylcholine ($^{18}$F-FCho).

data are due to failed PET tracer synthesis, paravenous injection or hardware problems. A total of 132 PET scans were included for analysis.

The ratio of the $SUV_{max}$ and $TBR_{max}$ post-therapy (day 2-5-9-12 for [$^{18}$F]FDG and day 1-6-8-13 for [$^{18}$F]FCho PET) to the $SUV_{max}$ and $TBR_{max}$ pre-therapy was not significantly different between the control group and the treated group, at any time point for both [$^{18}$F]FDG, at conventional and delayed time point, and for [$^{18}$F]FCho PET (S1 Fig).

To eliminate the influence of the differences in tumor volumes between individual animals, also MTV values were normalized to the MTV pre-therapy. Evolution of the normalized MTV and ($SUV_{mean}$ x normalized MTV) for [$^{18}$F]FDG (early and delayed) and [$^{18}$F]FCho PET are shown in Fig 4. Significant differences between control and therapy group are marked with an asterisk and listed in Table 2.

The MTV on conventional [$^{18}$F]FDG PET was significantly different between both groups on day 5 (p = 0.016). Using delayed [$^{18}$F]FDG PET imaging, significant differences in MTV

were present between both groups on day 9 (p = 0.032) and 12 (p = 0.032). No significant MTV differences were found between control and therapy group for [18F]FCho PET at any time point.

For conventional [18F]FDG PET, (SUV$_{mean}$ x normalized MTV) was significantly different between the control and treated group on day 5 (p = 0.008) post-irradiation using the last time frame of the dynamic PET acquisition. On delayed [18F]FDG PET a significant difference (SUV$_{mean}$ x normalized MTV) was found on day 2 (p = 0.032), day 9 (p = 0.032) and day 12 (p = 0.016) post-irradiation. No significant (SUV$_{mean}$ x normalized MTV) differences were found between control and treated group for [18F]FCho PET at any time point.

In Fig 5, tumor growth is clearly visible on contrast-enhanced T1-weighted MRI, conventional [18F]FDG, delayed [18F]FDG and [18F]FCho PET. Evolution of the normalized MTV and (SUV$_{mean}$ x normalized MTV) for [18F]FDG (conventional and delayed) and [18F]FCho PET in a rat receiving control treatment are shown in Fig 4. Significant differences between control and treatment group are marked with an asterisk and listed in Table 2.

The autoradiography image showed high [18F]FCho uptake in the F98 GB tumor and very low uptake in normal brain. The background corrected mean tumor-to-mean normal brain ratio was 3,72 and the max tumor-to-mean normal brain ratio was 6.84 (Fig 6A). The [18F]FCho uptake was clearly lower in the necrotic center of the F98 GB tumor (Fig 6B). EB (Fig 6C) and TRITC fluorescent images (Fig 6D) visualize the extravasation, confirming blood brain barrier breakdown. The extravasation is mainly present at the peritumoral edge and less in the necrotic center of the F98 GB tumor.

Fig 7 shows a comparison of contrast-enhanced T1-weighted MRI and [18F]FCho PET of GB patients compared to the F98 GB rat model. The contrast-enhanced T1-weighted MRI of patient (A) shows clearly necrosis in the tumor core, while this is not present in patient (B). However, both show a heterogeneous [18F]FCho uptake ranging from moderate to moderate intense (compared to the uptake in the scalp which serves as the reference) at the invasion front of the tumor (25–30 min post-injection; 380.7 MBq (A) and 372.0 MBq (B) injected activity). Low uptake is noted in the normal brain tissue. In the F98 GB rat tumor, no gross central tumor necrosis is seen on contrast-enhanced T1-weighted MRI (C) and increased [18F]

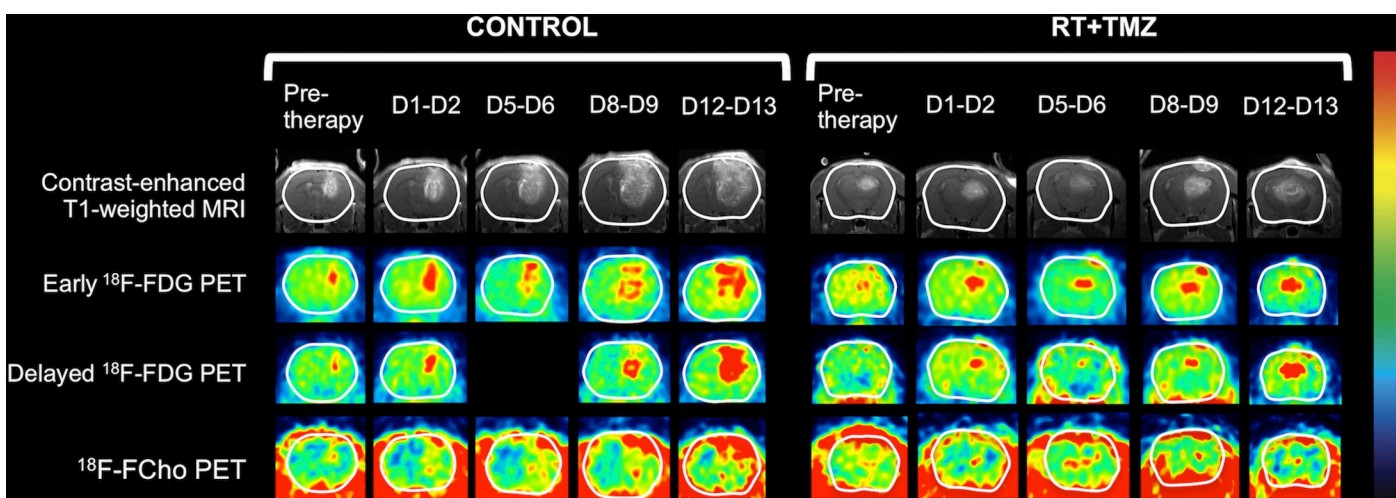

**Fig 5. Longitudinal follow-up on contrast-enhanced T1-weighted MRI (top row), conventional [18F]FDG PET (second row), delayed [18F]FDG (third row) and [18F]FCho PET images (bottom row) of control (left) and therapy (right) F98 GB rats.** For clarity, the brain is contoured in white, [18F]FDG PET (38.1 ± 0.6 MBq, 40–60 min post-injection and 240 min post-injection) and [18F]FCho PET (40.5 ± 0.7 MBq, 10–20 min post-injection) (mean ± SE). Data from one delayed [18F]FDG PET scan on day 5 is missing. Inhalational anesthesia with isoflurane (2%) mixed with oxygen (0.3 L/min) was used.

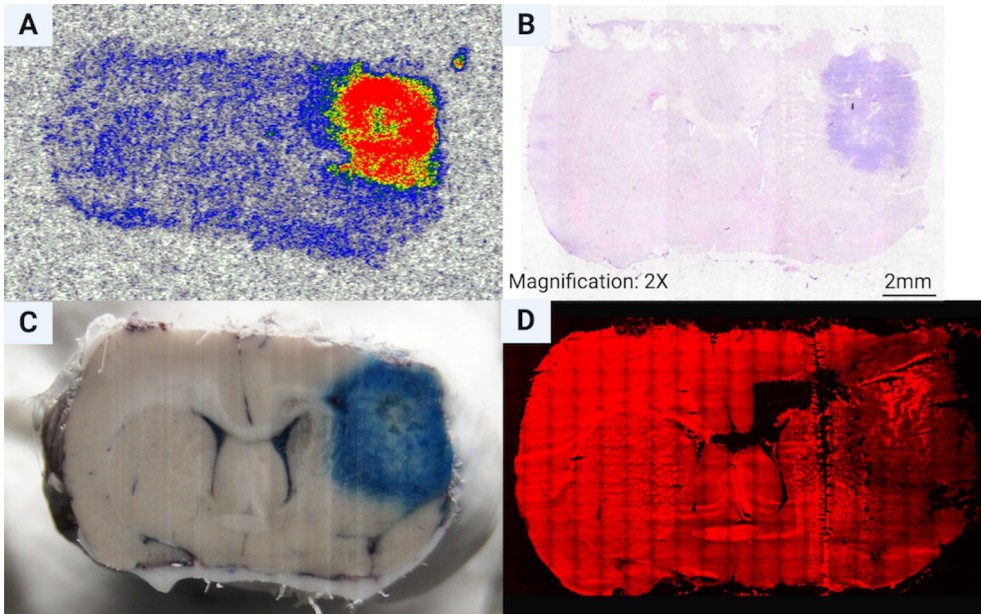

**Fig 6. Ex-vivo analysis of brain tissue sections.** (A) Autoradiography image with clear [18F]FCho uptake in the F98 GB tumor and a high tumor-to-normal brain tissue ratio (20 μm tissue section nr x) (B) Hematoxylin & Eosin staining of a consecutive slice (20 μm tissue section nr x+2) (C) Cryosected rat brain 1 hour after intravenous injection of Evans Blue (EB) and (D) TRITC fluorescent images visualize the extravasation, confirming blood brain barrier breakdown. The extravasation is mainly present at the peritumoral edge and less in the necrotic center of the F98 GB tumor (20 μm tissue section nr x+13).

FCho uptake is present only in the upper left margin of the tumor (20 min post injection; 39.6 MBq injected activity). Surrounding extra-cranial organs, such as the salivary glands and the masticatory muscles, show intense [18F]FCho uptake (D). This is clearly visible on the pre-clinical PET, while in humans this uptake is not visible within the axial brain slice. In (B) and (C-D), the leakage pattern of the gadolinium contrast agent on MRI differs strongly from the [18F]FCho uptake pattern. Diffuse leakage of Gd in the entire tumor volume is seen on MRI (C), while a more localized choline uptake is seen in the upper left margin of the tumor just beneath the skull (D).

## Discussion

We previously proposed an experimental rat model for MRI-guided conformal multiple arc GB treatment with a close resemblance to the image-guided conformal RT in the clinic with regard to beam usage. Irradiation with the SARRP and concomitant TMZ resulted in a stable tumor volume on serial MRI until nine days post-treatment, while continuous tumor growth was observed in the control group [43]. In this study, this optimized methodology was applied to investigate whether PET was able to detect treatment response earlier than contrast-enhanced MRI. For PET image analysis, an automatic threshold technique was chosen because it is known to be the best guarantee that consistent VOIs are defined on repeat PET scans [48–50]. Hence, for [18F]FDG and [18F]FCho PET, a threshold of $\geq$ 60% and $\geq$ 50% of the maximum uptake, respectively, was selected based on visual inspection (Fig 3). Based on our results, $SUV_{max}$ and $TBR_{max}$ were not able to detect any treatment effects at the chosen time points, nor using [18F]FDG nor [18F]FCho as a PET biomarker. However, we found that for [18F]FDG PET at conventional and delayed acquisition times, the parameter ($SUV_{mean}$ x MTV) was superior to MTV alone in detecting early treatment effect (Table 2). This is in agreement with

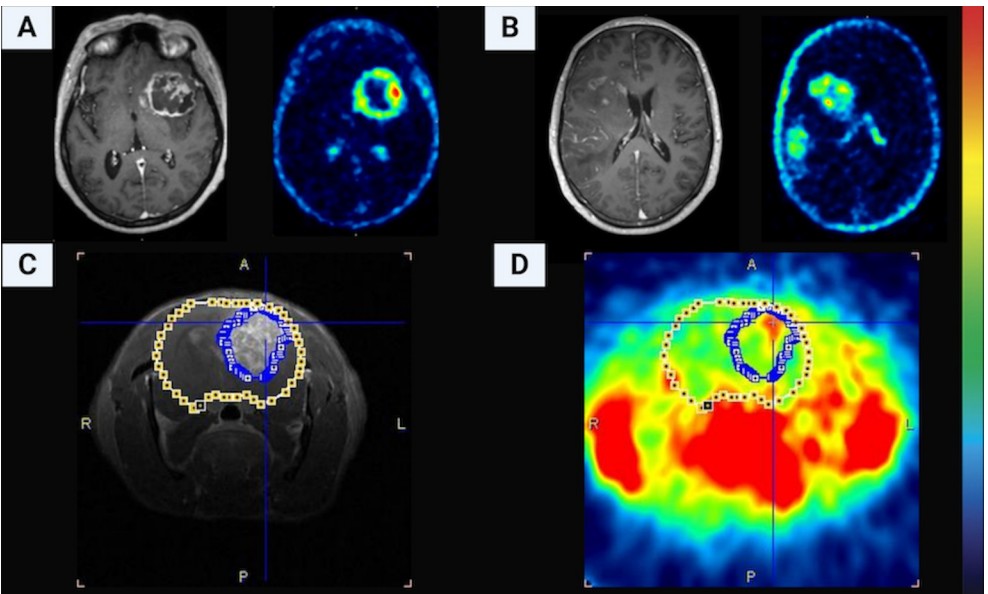

**Fig 7. Comparison of contrast-enhanced T1-weighted MRI and [18F]FCho PET of GB patients compared to the F98 GB rat model.** The contrast-enhanced T1-weighted MRI of patient (A) shows clearly necrosis in the tumor core, while this is not present in patient (B). Both patients show a heterogeneous [18F]FCho uptake ranging from moderate to moderate intense (compared to the uptake in the scalp which serves as the reference) at the invasion front of the tumor. Low uptake is noted in the normal brain tissue. In the F98 GB rat tumor, no gross central tumor necrosis is seen on contrast-enhanced T1-weighted MRI (C) and increased [18F]FCho uptake is present only in the upper left margin of the tumor (D).

our previous results in GB patients in which we found that both MTV and mean tracer uptake have to be taken into account to be able to capture therapy effects [48]. In case of [18F]FDG, the (SUV$_{mean}$ x MTV) is also referred to as total lesion glycolysis (TLG), which is a well-known volumetric parameter that enables to capture the glycolytic phenotype and overall tumor burden. In this study, using (SUV$_{mean}$ x MTV), [18F]FDG PET acquired 40–60 minutes post-injection, was able to detect treatment response as early as 5 days post-therapy. Similar results were found when evaluating the changes in contrast-enhanced tumor volume on MRI. Importantly, [18F]FDG PET acquired 4 hours post-injection was able to detect the treatment response even earlier, namely at day 2 post-irradiation (Table 2). This is in agreement with the advantages of dual time point FDG PET that were previously described [26, 27]. In Fig 4 can be observed that tumor volumes measured using contrast-enhanced MRI clearly increase over time in the control group, while metabolic tumor volumes remain more or less stable and also show larger variability. Our hypothesis is that the fast growing tumors, as observed on MRI, are becoming metabolically more heterogeneous tumors. This results in a more heterogeneous tracer uptake within these tumors (e.g. necrotic core vs. viable tumor proliferation and infiltration), which gives rise to higher variations of the measured MTV-values in the control group and that might also explain why the MTV-values are not increasing over time.

For those centers having access to amino-acid PET tracers, [11C]Methionine ([11C]MET) and [18F]FET PET have been suggested to be better suited than [18F]FDG for brain tumor imaging and monitoring therapy response in brain tumor patients [3]. Data in the literature suggest that a reduction of amino acid uptake by glioma is a sign of a favorable treatment response and a decreased tracer uptake as early as 7–10 days after the completion of treatment has been documented [15, 51–53]. [18F]FCho PET, first introduced for PET imaging of brain tumors by DeGrado *et al.* [36, 54, 55], has previously been investigated for therapy response

assessment in glioma, but only a few studies are available [56–58]. Li *et al.* reported that, for [$^{11}$C]Choline PET, a tumor-to-normal-brain ratio (TBR) ≤ 1.4 might predict a longer overall survival in patients with suspected recurrent glioma after treatment [56]. Parashar *et al.* suggested that there was a good correlation between a change in SUV$_{max}$ of the tumor volume during RT and response [57]. However, in the latter study, only one patient with a high-grade glioma was included. Finally, a [$^{18}$F]FCho PET study in childhood astrocytic tumors confirmed the added value of [$^{18}$F]FCho SUV$_{max}$ and functional MRI apparent diffusion coefficient values to monitor therapy response [58].

Our hypothesis is that [$^{18}$F]FCho PET might be able to detect a treatment-induced diminished cell proliferation rate because this choline PET analogue is a substrate for choline kinase, an enzyme commonly overexpressed in malignant lesions involved in the incorporation of choline into phospholipids, which is an essential component of all cell membranes. In cancer, an increased cellular transport and higher expression of choline kinase leads to an increased uptake of radiolabeled choline [34, 59]. In addition, the major advance for brain tumor imaging is that the choline uptake observed in normal cortex is only corresponding to 10% of the uptake registered with [$^{18}$F]FDG, ameliorating the delineation of tumor boundaries [60]. In a previous clinical study, we investigated the potential of [$^{18}$F]FCho PET compared to state-of-the art conventional MRI using RANO criteria for early therapy response assessment in GB patients. We found that SUV values were not able to predict response, while (SUV$_{mean}$ x MTV) allowed prediction of therapy response one month after the completion of radiation therapy, however, not earlier than changes of tumor volume derived from contrast-enhanced MRI [48].

In this study, we did not find significant differences at any time point for MTV and (SUV$_{mean}$ x MTV) of [$^{18}$F]FCho PET between the control and the treatment group. Based on these results, in rats, [$^{18}$F]FCho PET was not able to detect early combined radiation and chemotherapy effects after the completion of treatment. We can only speculate about an explanation.

First, it is worth mentioning that the time scale in humans and rats is quite different. In patients, early treatment response after concomitant chemo-radiation therapy is arbitrarily defined at 1 month after the completion of radiotherapy. In rats, we investigated post-treatment changes as early as a few days up to 2 weeks after the combined treatment. Moreover, it should be noted that in rats a single radiation was applied, whereas radiation treatment is fractionated by default in patients. Thus, it may well be that both the difference in time scale as well as the use of fractionated versus non-fractionated radiation treatment underlies the alleged conflicting findings in both studies.

Secondly, we hypothesize that differences in metabolism of [$^{18}$F]FCho between rat and humans play a major role in clarifying the results. This can only further be elucidated by applying a full kinetic modeling study [44, 45, 61–64]. However, such studies require an arterial blood input curve. Arterial blood sampling in patients is no longer approved by our local ethics committee. On the other hand, it was approved by our local preclinical ethics committee. Hence, we managed to obtain arterial blood sampling in F98 GB rats. We found an optimal fitting of [$^{18}$F]FCho uptake in the tumor using a reversible model, see Fig 8 with data of our previous study [45]. This was not in agreement with the two-compartmental model that we expected. Indeed, relying on the Kennedy pathway of choline metabolism, illustrating why choline may work as a PET tracer to visualize malignancies, a 2-compartmental model could be expected. Given the result of a reversible model as the best solution for our measures, we were not able to differentiate [$^{18}$F]FCho uptake in the tumor mediated by leakage through the damaged BBB from intracellular metabolic trapping. Choline is in addition rapidly oxidized in the mitochondria of liver and kidney to betaine. As a result, our data showed that [$^{18}$F]FCho in plasma decreased rapidly while a hydrophilic metabolite, most likely [$^{18}$F]Fluorobetaine,

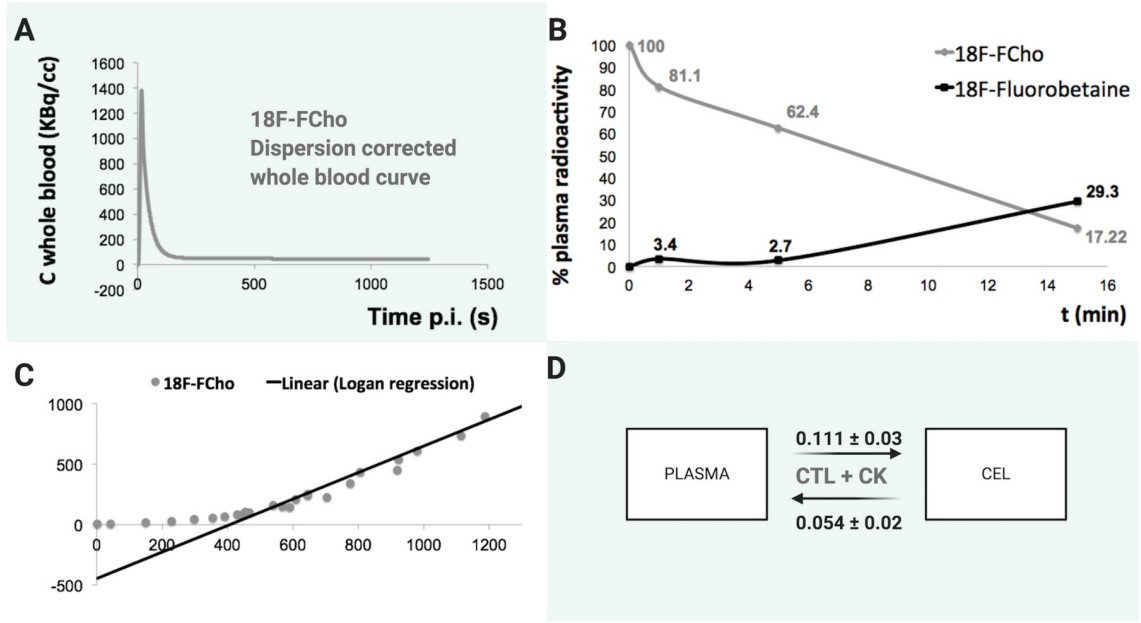

**Fig 8. Kinetic modeling of [¹⁸F]FCho in F98 GB rat model.** The fraction of plasma radioactivity contributing to [¹⁸F]FCho decreased rapidly to 17.2% concomitant with the appearance of a metabolite, most likely [¹⁸F]Fluorobetaine (B). Kinetic modeling and graphical analysis of [¹⁸F]FCho reached optimal fitting using a reversible model (C, D). Choline transporter (CTL), Choline kinase (CK). Data by Bolcaen *et al.*, previously published under CC BY 4.0 licence [45].

rapidly increased, see Fig 8B [45, 61]. In contrast, recently, the percentage of activity due to [¹⁸F]FCho in plasma of patients was more stable than in rats: 67±11%, 65±9%, 65±7% and 64 ±7% at 1, 5, 10 and 30 min after injection [65]. This rapid and extensive clearance from the blood after intravenous injection in rats makes an acquisition starting early after injection of [¹⁸F]FCho preferable. An 'early' [¹⁸F]FCho acquisition has also been suggested in the clinic. In prostate cancer patients, the biodistribution of [¹⁸F]FCho changed very slowly after 10-min post-injection and the activity in the prostate (with malignant involvement) reached a maximum within a 5-min window following the injection [54, 66]. For brain [¹⁸F]FCho, several clinical reports performed emission scanning for 15 min, beginning 5–10 min after injection of the tracer [36, 67]. Our group previously confirmed that [¹⁸F]FCho uptake by all types of brain lesions was rapid with minimal changes in uptake activity more than 6 min after administration, except for meningiomas [68]. In brain metastasis, intra-tumor [¹⁸F]FCho uptake also reached 80% and 90% of the total activity at 3±4 and 7±6 minutes post injection, respectively [65]. This confirms other studies that radiolabeled choline uptake is rapid and appears to reach a plateau faster than [¹⁸F]FET [69–72]. Importantly, delayed imaging is recommended if discrimination between meningioma and other brain tumors is of concern or for the detection of bone metastasis in prostate cancer patients [68, 73, 74].

## Conclusion

Based on a preclinical rat model for GB and multimodal imaging using MRI and PET with two different tracers, to evaluate early treatment response after combined chemo-radiation therapy, we found that both MRI and PET can be used for this purpose. With regard to the choice of PET biomarker, [¹⁸F]FDG (and particularly acquired 4 hours post-injection) is preferred over [¹⁸F]FCho. Further comparative studies should corroborate these results and should also include (different) amino acid PET tracers.

## Supporting information

**S1 Fig. The ratio of the $SUV_{max}$ and $TBR_{max}$ post-therapy to the $SUV_{max}$ and $TBR_{max}$ pre-therapy for both [$^{18}$F]FDG, at conventional and delayed time point, and for [$^{18}$F]FCho PET.**
(TIF)

## Author Contributions

**Conceptualization:** Julie Bolcaen, Benedicte Descamps, Filip De Vos, Christian Vanhove, Ingeborg Goethals.

**Data curation:** Christian Vanhove.

**Formal analysis:** Julie Bolcaen, Benedicte Descamps, Christian Vanhove.

**Funding acquisition:** Christian Vanhove, Ingeborg Goethals.

**Investigation:** Julie Bolcaen, Benedicte Descamps, Giorgio Hallaert, Christian Vanhove.

**Methodology:** Julie Bolcaen, Benedicte Descamps, Karel Deblaere, Tom Boterberg, Christian Vanhove.

**Project administration:** Julie Bolcaen, Benedicte Descamps, Christian Vanhove, Ingeborg Goethals.

**Resources:** Julie Bolcaen, Benedicte Descamps, Filip De Vos, Caroline Van den Broecke, Christian Vanhove, Ingeborg Goethals.

**Software:** Christian Vanhove.

**Supervision:** Christian Vanhove, Ingeborg Goethals.

**Validation:** Benedicte Descamps, Karel Deblaere, Filip De Vos, Tom Boterberg, Giorgio Hallaert, Caroline Van den Broecke, Christian Vanhove, Ingeborg Goethals.

**Visualization:** Julie Bolcaen, Christian Vanhove, Ingeborg Goethals.

**Writing – original draft:** Julie Bolcaen, Benedicte Descamps.

**Writing – review & editing:** Karel Deblaere, Filip De Vos, Tom Boterberg, Giorgio Hallaert, Caroline Van den Broecke, Christian Vanhove, Ingeborg Goethals.

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
