## [Decision Letter · Decision Letter 0]

5 Jan 2021

PONE-D-20-36019

Assessment of the effect of therapy in a rat model of glioblastoma using [18F]FDG and [18F]FCho PET compared to contrast-enhanced MRI.

PLOS ONE

Dear Dr. Bolcaen,

Thank you for submitting your manuscript to PLOS ONE. After careful consideration, we feel that it has merit but does not fully meet PLOS ONE’s publication criteria as it currently stands. Therefore, we invite you to submit a revised version of the manuscript that addresses the points raised during the review process.

The article describes the results of a pre-clinical study in a rat model of glioblastoma using [18F]FDG and [18F]FCho PET compared to contrast-enhanced MRI"  for the early detection of treatment response. The objective of the study is interesting and may help future potential applications regard PET imaging in the field of primary brain tumors. However, the paper needs a revision as defined in the section of comments below.

We look forward to receiving your revised manuscript.

Kind regards,

Pierpaolo Alongi

Academic Editor

PLOS ONE

Additional Editor Comments:

The article describes the results of a pre-clinical study in a rat model of glioblastoma using [18F]FDG and [18F]FCho PET compared to contrast-enhanced MRI" for the early detection of treatment response. The objective of the study is interesting and may help future potential applications regard PET imaging in the field of primary brain tumors. I agree with the comments of the two reviewers the paper needs a major revision.

The results of choline PET and FDG PET have to be discussed carefully because both radiopharmaceutical agents have limited use in this field. For choline PET, despite the biodistribution of the tracer is fast compared to FDG other studies suggest a start acquisition rapidly after injection and a time between 20 and 50 minutes for late imaging in order to have a good balance of T/B ratio. A unique Cho-PET dynamic acquisition 5-20 minutes after the injection may affect negatively the quality of the images. Please discuss in the discussion and eventually in the limitation of the study.

I suggest also to report some references missing of recent representative articles on humans: Eg.

- Vetrano, I.G., Laudicella, R. & Alongi, P. Choline PET/CT and intraoperative management of primary brain tumors. New insights for contemporary neurosurgery. Clin Transl Imaging 8, 401–404 (2020). https://doi.org/10.1007/s40336-020-00398-6

- Alongi, P., Quartuccio, N., Arnone, A. et al. Brain PET/CT using prostate cancer radiopharmaceutical agents in the evaluation of gliomas. Clin Transl Imaging 8, 433–448 (2020). https://doi.org/10.1007/s40336-020-00389-7

- Fraioli F, Shankar A, Hargrave D, Hyare H, Gaze MN, Groves AM, Alongi P, Stoneham S, Michopoulou S, Syed R, Bomanji JB. 18F-fluoroethylcholine (18F-Cho) PET/MRI functional parameters in pediatric astrocytic brain tumors. Clin Nucl Med. 2015 Jan;40(1):e40-5. doi: 10.1097/RLU.0000000000000556. PMID: 25188640.

Alongi P, Vetrano IG, Fiasconaro E, Alaimo V, Laudicella R, Bellavia M, Rubino F, Bagnato S, Galardi G. Choline-PET/CT in the Differential Diagnosis Between Cystic Glioblastoma and Intraparenchymal Hemorrhage. Curr Radiopharm. 2019;12(1):88-92. doi: 10.2174/1874471011666180817122427. PMID: 30117406.

2. We note that Figure 8 in your submission contains copyrighted images. All PLOS content is published under the Creative Commons Attribution License (CC BY 4.0), which means that the manuscript, images, and Supporting Information files will be freely available online, and any third party is permitted to access, download, copy, distribute, and use these materials in any way, even commercially, with proper attribution. For more information, see our copyright guidelines: http://journals.plos.org/plosone/s/licenses-and-copyright.

(1) You may seek permission from the original copyright holder of Figure(s) [#] to publish the content specifically under the CC BY 4.0 license.

Reviewers' comments:

Reviewer's Responses to Questions

**Comments to the Author**

1. Is the manuscript technically sound, and do the data support the conclusions?

Reviewer #1: Yes

Reviewer #2: Partly

2. Has the statistical analysis been performed appropriately and rigorously? 

Reviewer #1: Yes

Reviewer #2: No

3. Have the authors made all data underlying the findings in their manuscript fully available?

Reviewer #1: Yes

Reviewer #2: Yes

4. Is the manuscript presented in an intelligible fashion and written in standard English?

Reviewer #1: Yes

Reviewer #2: Yes

5. Review Comments to the Author

Reviewer #1: In this study, the authors investigated the role of FDG and F-Choline PET compared to MRI for the early detection of treatment response in a glioblastoma rat model obtained with F98 cells. Rats were divided into two groups (control and treated with radiation and Temozolomide) and the response was monitored with MRI and PET with FDG and F-Choline performed at different times points.

The text is well organized and methods and results are fully described but there are some points that need to be clarified.

Major comments

Abstract

1. Line 43, in the abstract authors indicate that F-Choline was performed on day 7 post-treatment but in M&M and in the text is indicated day 8 (line 218).

Materials and methods

1. Line 231, authors can edit the correct injected dose (mean ± SE) of FDG and F-Choline because in line 231 authors indicated 37 MBq and in the figure legends (Fig.3 and Fig. 5) authors indicated other specific doses.

2. Line 274, the dose of FCho can be edited in MBq, please?

3. Lines- 289-292, why these lines are under the paragraph “Autoradiography and Evans Blue (EB) staining”, can authors add another title, please?

4. Figure 3, is the same animal? Because the images are different. If not, can authors use the same animal, please?

Results

5. In table 2 there are only the p-values, can also add the value of each parameter (mean ± SE), please?

6. Figure 4, why FDG MTV values at d2 are so different between control and treated group whereas the volume measured using MRI is closer? The tumor volume of control animals measured using MRI increased along time whereas MTV (both FDG and Choline) remained stable or slight decrease, what is the hypothesis? Authors should discuss.

7. On day 9, only 2 control animals performed FDG PET, how is it possible that both MTV and MTV+SUVmean values are significantly different between control and treated group (line 347 and 353)?

8. In figure 5 there is represented only a control rat, authors can add a longitudinal figure with a representative treated rat so it is possible compared images of control and treated rats, please?

9. Figure 5, what is the color scale for PET (SUV, %ID/g, TBR)? Can authors also add min and max values on the scale.

10. Did authors evaluate post mortem staining for ki67, GFAP, choline kinase?

Discussion

Edit discuss on the basis of results (point 6), please.

Reviewer #2: The authors evaluate the role of FDG-PET and Cho-PET, compared to c.e. MRI, for the early detection of treatment response in murine model of GBM; 5 animals randomizedly received RT plus TMZ, while other 5 no. The treatment effect was evaluated with serial MRI and FDG-PET (day 2, 5, 9, 12 post-treatment), and also Cho-PET (day 1, 6, 7, 13). The metabolic tumor volume (MTV) was semi-automatically calculatedm and the average tracer uptake within the MTV was converted to a SUV. Using SUVmean x MTV, FDG-PET started to detect treatment's effects at 5 day post-treatment, comparable to c.e. MRI. Moreover, delayed FDG-PET (240 min p.i.) earlier detect such effects (from day 2); on the other hand, no significant differences were found at any time point for both the MTV and (SUVmean x MTV) of Cho-PET. Therefore, the authors concluded that MRI and delayed FDG-PET detect early treatment responses in GB murine model of GBM, whereas these results were not obtained with Cho-PET

The topic is undoubtedly intriguing, but i have some issues:

-INTRODUCTION

The ref 1 is related to 2007 WHO classification; from an epidemiological point of view, it should be better to consider the last CBTRUS report (Ostrom et al., CBTRUS Statistical Report: Primary Brain and Other Central Nervous System Tumors Diagnosed in the United States in 2013–2017, Neuro-Oncology, Volume 22, Supplement 1, October 2020).

I suggest also to modify ref. 2 and 3, using a more up-to-date literature reference about glioma management (i.e., Weller et al., European Association for Neuro-Oncology (EANO) guideline on the diagnosis and treatment of adult astrocytic and oligodendroglial gliomas. Lancet Oncol. 2017 Jun;18(6):e315-e329).

Moreover, the study by Stupp in 2005 that showed the role of combined RT-CMT was not the ref n° 5 but the one published in NEJM (Stupp et al, Radiotherapy plus concomitant and adjuvant temozolomide for glioblastoma. N Engl J Med. 2005 Mar 10;352(10):987-96. doi: 10.1056/NEJMoa0433309).

It shuld be better to update the references related to the clinical role of ChoPET in brain tumors (the authors cite a quite old ref, the number 35, which was a review of literature available more than 10 years ago), due to the increasing interest about such technique.

Why the authors selected Cho-PET, instead of [18F]FAZA PET, for example? I think that clarifying the advantages and disadvantages os this choice could increase the informative role of the present work.

How where the simple size selected?statistical analysis were performed to selected a population of 10 animals?

Finally, the author disclosure a financial support by Lux Luka Foundation, but they must clearly state, according to Journal guidelines, who exactly received fundings, and the role of the sponsor in the study design and analysis.

6. PLOS authors have the option to publish the peer review history of their article (what does this mean?). If published, this will include your full peer review and any attached files.

Reviewer #1: No

Reviewer #2: No

---

## [Author Response · Author response to Decision Letter 0]

10 Feb 2021

A. Additional Editor Comments:

The article describes the results of a pre-clinical study in a rat model of glioblastoma using [18F]FDG and [18F]FCho PET compared to contrast-enhanced MRI" for the early detection of treatment response. The objective of the study is interesting and may help future potential applications regard PET imaging in the field of primary brain tumors. I agree with the comments of the two reviewers the paper needs a major revision.

Comment 1: The results of choline PET and FDG PET have to be discussed carefully because both radiopharmaceutical agents have limited use in this field. For choline PET, despite the biodistribution of the tracer is fast compared to FDG other studies suggest a start acquisition rapidly after injection and a time between 20 and 50 minutes for late imaging in order to have a good balance of T/B ratio. A unique Cho-PET dynamic acquisition 5-20 minutes after the injection may affect negatively the quality of the images. Please discuss in the discussion and eventually in the limitation of the study.

The main reason for starting the acquisition early (5–10 min) after injection of the tracer is indeed because of the rapid and extensive clearance from the blood after intravenous injection. DeGrado et al. documented that the biodistribution of [18F]FCho changes very slowly after 10-min post-injection (DeGrado TR 2001 and 2002). In 2001, in prostate cancer patients, it was documented that the activity in the prostate (with known malignant involvement) reached a maximum within a 5-min window following the injection (DeGrado TR 2001). This was also confirmed by our group investigating the blood kinetics of [18F]FCho in rats, as shown in Figure 8, confirming a fast metabolization with an availability of only 17.5% of intact tracer after 15 min (Bolcaen et al. 2016). In multiple clinical reports, emission scanning of the brain was performed for 15 min, beginning 5–10 min after injection of the tracer (Kwee et al. 2004, 2007). To confirm this, in a previous publication of our group, we did investigate the optimal timing for imaging brain tumours and other brain lesions with [18F]FCho PET (Mertens et al. 2012). On the basis of the TACs, PET imaging with [18F]FCho starting within minutes after the administration of the tracer is preferred and uptake by all types of brain lesions was rapid, and minimal changes in uptake activity occurred more than 6 min after the administration of the tracer, except for meningiomas that showed decreasing activity after an early peak (Mertens K et al. 2012). Hence, if discrimination between meningioma and other brain tumours is of concern, both 'early' and 'late' PET imaging could be helpful. Recently, Grkovski M et al. performed a dynamic 40 min [18F]FCho PET in patients with brain metastasis. The percentage of activity due to [18F]FCho in plasma was more stable than in rats: 67±11%, 65±9%, 65±7% and 64±7% at 1, 5, 10 and 30 min after injection. However, intratumor [18F]FCho uptake reached 80% and 90% of the total activity at 3±4 and 7±6 minutes (median 1 and 6 minutes) post injection, respectively (Grkovski M 2020). This confirms other studies that radiolabeled choline uptake is rapid and sustained (Schaefferkoetter JD 2017, Grkovski M 2018, Sutinen E 2004) and appears to reach a plateau faster than [18F]FET (Lohmann P 2015).

In contrast with [18F]FDG brain imaging, where dual time point imaging has shown clear advantages to increase the T/B ratio (Mertens et al. 2013, Spence AM et al. 2004), this is assumed to be less advantageous for [18F]FCho PET since the uptake in normal brain is already low (Mertens K et al. 2012). For the use of [18F]FCho PET for the detection of bone metastasis in prostate cancer patients, delayed imaging is recommended. A significant increase in [18F]FCho accumulation in bone metastases was documented using dual-time-point PET imaging (Kwee et al. 2006, Husarik DB et al. 2008).

Based on the above-mentioned we selected a dynamic scan of 20 min for imaging the F98 GB tumor in rats assuming based on the literature that the maximal tumor uptake has been reached by then. Parts of this clarification were added to the discussion: line 567-583.

● Mertens K, Bolcaen J, Ham H, Deblaere K, Van den Broecke C, Boterberg T, De Vos F, Goethals I. The optimal timing for imaging brain tumours and other brain lesions with 18F-labelled fluoromethylcholine: a dynamic positron emission tomography study. Nucl Med Commun. 2012;33:954-9.

● Mertens K, Ham H, Deblaere K, Kalala JP, Van den Broecke C, Slaets D, et al. Distribution patterns of 18F-labelled fluoromethylcholine in normal structures and tumors of the head: a PET/MRI evaluation. Clin Nucl Med. 2012;37:e196-203.

● Mertens K, Acou M, Van Hauwe J, De Ruyck I, Van den Broecke C, Kalala JP, et al. Validation of 18F-FDG PET at conventional and delayed intervals for the discrimination of high-grade from low-grade gliomas: a stereotactic PET and MRI study. Clin Nucl Med. 2013;38:495-500. 

● Spence AM, Muzi M, Mankoff DA, O’Sullivan SF, Link JM, Lewellen TK, et al. 18F-FDG PET of gliomas at delayed intervals: improved distinction between tumor and normal gray matter. J Nucl Med. 2004;45:1653-9.

● DeGrado TR, Coleman RE, Wang S, Baldwin SW, Orr MD, Robertson CN, et al. Synthesis and evaluation of 18F-labeled choline as an oncologic tracer for positron emission tomography: initial findings in prostate cancer. Cancer Res 2001;61:110-7.

● DeGrado TR, Reiman RE, Price DT, Wang S, Coleman RE. Pharmacokinetics and radiation dosimetry of 18F-fluorocholine. J Nucl Med. 2002;43:92-6.

● Kwee SA, Wei H, Sesterhenn I, Yun D, Coel MN. Localization of primary prostate cancer with dual-phase 18F-fluorocholine PET. J Nucl Med 2006;47:262-9.

● Kwee SA, Coel MN, Lim J, Ko JP. Combined use of F-18 fluorocholine positron emission tomography and magnetic resonance spectroscopy for brain tumor evaluation. J Neuroimaging. 2004;14:285-9.

● Kwee SA, Ko JP, Jiang CS, Watters MR, Coel MN. Solitary brain lesions enhancing at MR imaging: evaluation with fluorine 18fluorocholine PET. Radiology. 2007;244:557-65.

● Husarik DB, Miralbell R, Dubs M, John H, Giger OT, Gelet A, et al. Evaluation of [(18)F]-choline PET/CT for staging and restaging of prostate cancer. Eur J Nucl Med Mol Imaging. 2008;35:253-63.

● Grkovski M, Kohutek ZA, Schöder H, Brennan CW, Tabar VS, Gutin PH, et al. 18F-Fluorocholine PET uptake correlates with pathologic evidence of recurrent tumor after stereotactic radiosurgery for brain metastases. Eur J Nucl Med Mol Imaging. 2020;47:1446-57.

● Schaefferkoetter JD, Wang Z, Stephenson MC, Roy S, Conti M, Eriksson L, et al. Quantitative 18F-fluorocholine positron emission tomography for prostate cancer: correlation between kinetic parameters and Gleason scoring. EJNMMI Res. 2017;7:25.

● Grkovski M, Gharzeddine K, Sawan P, Schöder H, Michaud L, Weber WA, et al. 11C-Choline Pharmacokinetics in Recurrent Prostate Cancer. J Nucl Med. 2018;59:1672-8.

● Sutinen E, Nurmi M, Roivainen A, Varpula M, Tolvanen T, Lehikoinen P, et al. Kinetics of [(11)C]choline uptake in prostate cancer: a PET study. Eur J Nucl Med Mol Imaging. 2004;31:317-24.

● Lohmann P, Herzog H, Rota Kops E, Stoffels G, Judov N, Filss C, et al. Dual-time-point O-(2-[(18)F]fluoroethyl)-L-tyrosine PET for grading of cerebral gliomas. Eur Radiol. 2015;25:3017-24. 

● Bolcaen J, Lybaert K, Moerman L, Descamps B, Deblaere K, Boterberg T, et al. Kinetic Modeling and Graphical Analysis of 18F-Fluoromethylcholine (FCho), 18F-Fluoroethyltyrosine (FET) and 18F-fluorodeoxyglucose (FDG) PET for the Discrimination between High-grade Glioma and Radiation Necrosis in Rats. PLoS One. 2016;11:e0161845.

Comment 2: I suggest also to report some references missing of recent representative articles on humans: Eg.

- Vetrano, I.G., Laudicella, R. & Alongi, P. Choline PET/CT and intraoperative management of primary brain tumors. New insights for contemporary neurosurgery. Clin Transl Imaging 8, 401–4 (2020). https://doi.org/10.1007/s40336-020-00398-6

- Alongi, P., Quartuccio, N., Arnone, A. et al. Brain PET/CT using prostate cancer radiopharmaceutical agents in the evaluation of gliomas. Clin Transl Imaging 8, 433–48 (2020). https://doi.org/10.1007/s40336-020-00389-7

- Fraioli F, Shankar A, Hargrave D, Hyare H, Gaze MN, Groves AM, Alongi P, Stoneham S, Michopoulou S, Syed R, Bomanji JB. 18F-fluoroethylcholine (18F-Cho) PET/MRI functional parameters in pediatric astrocytic brain tumors. Clin Nucl Med. 2015 Jan;40(1):e40-5. doi: 10.1097/RLU.0000000000000556. PMID: 25188640.

-Alongi P, Vetrano IG, Fiasconaro E, Alaimo V, Laudicella R, Bellavia M, Rubino F, Bagnato S, Galardi G. Choline-PET/CT in the Differential Diagnosis Between Cystic Glioblastoma and Intraparenchymal Hemorrhage. Curr Radiopharm. 2019;12(1):88-92. doi: 10.2174/1874471011666180817122427. PMID: 30117406. 

We agree that these recent references are important and included these in the manuscript. This section was added to the introduction line 150-157:

The metabolic information acquired by [18F]FCho PET has been shown to be able to distinguish high-grade glioma, brain metastases and benign lesions and to identify the most malignant areas for stereotactic sampling [Mertens et al. 2010, Kwee et al. 2007, Alongi et al. 2020, Vetrano et al. 2020]. Grech-Sollars et al. concluded that [18F]FCho PET was able to differentiate WHO (World Health Organization) grade IV from grade II and III tumours, whereas MR spectroscopy differentiated grade III/IV from grade II tumours [Grech-Sollars et al. 2019]. Recently, the potential use of [18F]FCho PET/CT in the intraoperative management or radio-surgical approaches for glioma has been suggested, including intraoperative guidance in conjunction with MR spectroscopy [Alongi et al. 2019, Vetrano et al. 2020, Villena Martin et al. 2020].

The reference to Fraioli et al. was added in the discussion Line 518 and 523-525: 

Finally, a [18F]FCho PET study in childhood astrocytic tumors confirmed the added value of [18F]FCho SUVmax and functional MRI apparent diffusion coefficient values to monitor therapy response [Fraioli et al. 2015].

Also these recent reference were included in the revised manuscript: 

- Villena Martín M, Pena Pardo FJ, Jiménez Aragón F, Borras Moreno JM, García Vicente AM, et al. Metabolic targeting can improve the efficiency of brain tumor biopsies. Semin Oncol. 2020;47:148-54.

- Grech-Sollars M, Ordidge KL, Vaqas B, Davies C, Vaja V, Honeyfield L, et al. Imaging and Tissue Biomarkers of Choline Metabolism in Diffuse Adult Glioma: 18F-Fluoromethylcholine PET/CT, Magnetic Resonance Spectroscopy, and Choline Kinase α. Cancers. 2019;11:1969.

B. When submitting your revision, we need you to address these additional requirements.

Comment 3: Please ensure that your manuscript meets PLOS ONE's style requirements, including those for file naming. The PLOS ONE style templates can be found at

Upon submission of the revised manuscript, extra care was taken to meet the PLOS ONE style requirements, including those for file naming. Referring to supplemental materials was adapted and author titels were deleted. The reference style of the added references was adapted in the final manuscript. Figures sizes were adapted to meet the criteria.

Comment 4:

We note that Figure 8 in your submission contains copyrighted images. All PLOS content is published under the Creative Commons Attribution License (CC BY 4.0), which means that the manuscript, images, and Supporting Information files will be freely available online, and any third party is permitted to access, download, copy, distribute, and use these materials in any way, even commercially, with proper attribution. For more information, see our copyright guidelines: http://journals.plos.org/plosone/s/licenses-and-copyright.

(1) You may seek permission from the original copyright holder of Figure(s) [#] to publish the content specifically under the CC BY 4.0 license.

Figure 8 is published in our previous PLoS One publication: 2016;11(8):e0161845, Fig S2 and S4. This content is published under CC BY 4.0 license. To the best of our knowledge, an additional permission request is not required. Please let us know if we understood wrong. We changed the caption to clarify the reuse of the figure (line 589-590). The original figures were uploaded as ‘other’ in the online submission. 

Comment 5: Please include captions for your Supporting Information files at the end of your manuscript, and update any in-text citations to match accordingly. Please see our Supporting Information guidelines for more information: http://journals.plos.org/plosone/s/supporting-information.

The guidelines were applied for the supporting information, including an in-text citation: S1 Fig.

The name of the supporting information figure was matched with the supporting information captions within the manuscript (line 396). 

A caption was added at the end of the manuscript, including a title (line 819-820).

C. REVIEWER 1

In this study, the authors investigated the role of FDG and F-Choline PET compared to MRI for the early detection of treatment response in a glioblastoma rat model obtained with F98 cells. Rats were divided into two groups (control and treated with radiation and Temozolomide) and the response was monitored with MRI and PET with FDG and F-Choline performed at different times points.

The text is well organized and methods and results are fully described but there are some points that need to be clarified.

Major comments

Abstract

Comment 1. Line 43, in the abstract authors indicate that F-Choline was performed on day 7 post-treatment but in M&M and in the text is indicated day 8 (line 218).

This is indeed an error in the abstract and has been modified (line 52). [18F]FCho PET was performed on day 1-6-8-13 as mentioned in M&M line 247 and in Fig 2, Table 1 and Table 2.

Comment 2. Line 231, authors can edit the correct injected dose (mean ± SE) of FDG and F-Choline because in line 231 authors indicated 37 MBq and in the figure legends (Fig.3 and Fig. 5) authors indicated other specific doses.-4

We agree to include the mean injected activity of all [18F]FDG and [18F]FCho scans in the M&M. The mean injected activity for all [18F]FDG scans was 37.89 ± 0.35 MBq and for all [18F]FCho scans it was 39.55 ± 0.37 MBq (mean ± SE). This was added to the manuscript line 263-264. 

Comment 3. Line 274, the dose of FCho can be edited in MBq, please?

The activity (0.55 mCi) was changed to MBq at line 313.

Comment 4. Lines- 289-292, why these lines are under the paragraph “Autoradiography and Evans Blue (EB) staining”, can authors add another title, please?

We agree that these lines do not fit under that paragraph. These lines were moved to lines 302-305. 

Comment 5. Figure 3, is the same animal? Because the images are different. If not, can authors use the same animal, please?

The axial PET images used in Figure 3 were indeed of 3 different rats. We made a new figure including coronal images of an [18F]FDG (early and late) and a [18F]FCho PET of the same rat with a F98 GB tumor. Different thresholds (≥40-50-60-70 %) are contoured. The figure legend was adapted (line 294-300).

Results

Comment 6. In table 2 there are only the p-values, can also add the value of each parameter (mean ± SE), please?

The mean ± SE was added to table 2 (page 13-14).

Comment 7. Figure 4, why FDG MTV values at d2 are so different between control and treated group whereas the volume measured using MRI is closer? The tumor volume of control animals measured using MRI increased along time whereas MTV (both FDG and Choline) remained stable or slight decrease, what is the hypothesis? Authors should discuss.

The following paragraph has been added to the Discussion (line 503-510):

In Figure 4 can be observed that tumor volumes measured using contrast-enhanced MRI clearly increase over time in the control group, while metabolic tumor volumes remain more or less stable and also show larger variability. Our hypothesis is that the fast growing tumors, as observed on MRI, are becoming metabolically more heterogeneous tumors. This results in a more heterogeneous tracer uptake within these tumours (e.g. necrotic core vs. viable tumour proliferation and infiltration), which gives rise to higher variations of the measured MTV-values in the control group and that might also explain why the MTV-values are not increasing over time.

In addition (not added to the manuscript):

For example, the FDG MTV at day 2 is indeed very different between the control and therapy group whereas the difference of the MRI Gd tumor volume is smaller. When taken a closer look to the data, this can be clarified because 2 rats of the DMSO group showed a 12x increase in FDG MTV between pre-therapy and d2, whereas there was only a 2 to 3 fold increase of the MR Gd volume. The 3 other rats of the DMSO group showed a 2 to 3 fold increase of both the MR Gd volume and FDG MTV between pre-therapy and d2. As a result, the SE-values are higher.

Comment 8. On day 9, only 2 control animals performed FDG PET, how is it possible that both MTV and MTV+SUVmean values are significantly different between control and treated group (line 347 and 353)?

Thank you for this very valuable comment. After re-checking the data and statistical analysis, we observed a mistake in Table 1. On day 9 FDG PET was performed on 4 control animals and not on 2 control animals. We changed this in the table on page 10 and the total scans included was also changed: line 391.

Comment 9. In figure 5 there is represented only a control rat, authors can add a longitudinal figure with a representative treated rat so it is possible compared images of control and treated rats, please?

Longitudinal FDG and FCho PET/MRI images of treated rats were added to figure 5. 

The figure legend was adapted. Mean ± SE injected activity was added of all FDG and FCho PET scans included in the new figure (see line 424-430).

Comment 10. Figure 5, what is the color scale for PET (SUV, %ID/g, TBR)? Can authors also add min and max values on the scale.

The images in Figure 5 were created using the PMOD software and the color scale of the images is in kBq/cc (kBq/mL) before extracting data. Only after VOI (MTV) delineation, the uptake values in kBq/cc were extracted in excel to calculate SUV and TBR values. 

The contrast range of the images selected in Figure 5 was (0-850 kBq/cc) for early FDG, (0-220 kBq/cc) for delayed FDG and (0-350 kBq/cc) for FCho. These were selected manually to obtain an optimal image of the brain and GB tumor uptake. Hence we prefer not to add the different ranges to the color scale.

10. Did authors evaluate post mortem staining for ki67, GFAP, choline kinase?

These stainings were not performed in this study. We did make a lot of efforts to optimize IHC for staining the choline transporter (CTL1) to correlate with the [18F]FCho PET uptake. However, after mulitple unsuccesfull attempts, the antibody we purchased seemed to only bind human brain tissue and not rat brain tissue (although it was assumed to work by the company).

Discussion

Edit discussion on the basis of results (point 6), please.

The following paragraph has been added to the Discussion (line 503-510).

D. REVIEWER 2

Reviewer #2: The authors evaluate the role of FDG-PET and Cho-PET, compared to c.e. MRI, for the early detection of treatment response in murine model of GBM; 5 animals randomizedly received RT plus TMZ, while other 5 no. The treatment effect was evaluated with serial MRI and FDG-PET (day 2, 5, 9, 12 post-treatment), and also Cho-PET (day 1, 6, 7, 13). The metabolic tumor volume (MTV) was semi-automatically calculatedm and the average tracer uptake within the MTV was converted to a SUV. Using SUVmean x MTV, FDG-PET started to detect treatment's effects at 5 day post-treatment, comparable to c.e. MRI. Moreover, delayed FDG-PET (240 min p.i.) earlier detect such effects (from day 2); on the other hand, no significant differences were found at any time point for both the MTV and (SUVmean x MTV) of Cho-PET. Therefore, the authors concluded that MRI and delayed FDG-PET detect early treatment responses in GB murine model of GBM, whereas these results were not obtained with Cho-PET

The topic is undoubtedly intriguing, but i have some issues:

-INTRODUCTION

1. The ref 1 is related to 2007 WHO classification; from an epidemiological point of view, it should be better to consider the last CBTRUS report (Ostrom et al., CBTRUS Statistical Report: Primary Brain and Other Central Nervous System Tumors Diagnosed in the United States in 2013–2017, Neuro-Oncology, Volume 22, Supplement 1, October 2020).

Reference 1 was changed to the recent publication of Ostrom et al. We included numbers from this work in the introduction, see line 72-74:

‘In the US, 84,170 new cases of primary brain and other central nervous system tumors are estimated to be diagnosed in 2021. Glioblastoma (GB) has the highest number of cases of all malignant tumors, with 12,970 cases projected in 2021 [1].’

2. I suggest also to modify ref. 2 and 3, using a more up-to-date literature reference about glioma management (i.e., Weller et al., European Association for Neuro-Oncology (EANO) guideline on the diagnosis and treatment of adult astrocytic and oligodendroglial gliomas. Lancet Oncol. 2017 Jun;18(6):e315-e329).

We agree that this is an important and more up to date reference, hence we changed previous references 2 and 3 to Weller et al. 2017

3. Moreover, the study by Stupp in 2005 that showed the role of combined RT-CMT was not the ref n° 5 but the one published in NEJM (Stupp et al, Radiotherapy plus concomitant and adjuvant temozolomide for glioblastoma. N Engl J Med. 2005 Mar 10;352(10):987-96. doi: 10.1056/NEJMoa0433309).

Reference 5 was changed to this reference of Stupp et al. from NEJM.

4. It should be better to update the references related to the clinical role of ChoPET in brain tumors (the authors cite a quite old ref, the number 35, which was a review of literature available more than 10 years ago), due to the increasing interest about such technique.

We added more recent references related to the clinical role of ChoPET in brain tumors to the manuscript. An extra section was added to the introduction, see line 150-157. The reference to Fraioli et al. was added in the discussion Line 518 and 523-525. We included the following references:

- Vetrano IG, Laudicella R, Alongi P. Choline PET/CT and intraoperative management of primary brain tumors. New insights for contemporary neurosurgery. Clin Transl Imaging. 2020;8:401-4. 

- Alongi, P, Quartuccio, N, Arnone, A, Kokomani A, Allocca M, Nappi G, et al. Brain PET/CT using prostate cancer radiopharmaceutical agents in the evaluation of gliomas. Clin Transl Imaging. 2020;8:433-48. 

- Fraioli F, Shankar A, Hargrave D, Hyare H, Gaze MN, Groves AM, et al. 18F-fluoroethylcholine (18F-Cho) PET/MRI functional parameters in pediatric astrocytic brain tumors. Clin Nucl Med. 2015;40:e40-5.

- Alongi P, Vetrano IG, Fiasconaro E, Alaimo V, Laudicella R, Bellavia M, et al. Choline-PET/CT in the Differential Diagnosis Between Cystic Glioblastoma and Intraparenchymal Hemorrhage. Curr Radiopharm. 2019;12:88-92. 

- Villena Martín M, Pena Pardo FJ, Jiménez Aragón F, Borras Moreno JM, García Vicente AM, et al. Metabolic targeting can improve the efficiency of brain tumor biopsies. Semin Oncol. 2020;47:148-54.

- Grech-Sollars M, Ordidge KL, Vaqas B, Davies C, Vaja V, Honeyfield L, et al. Imaging and Tissue Biomarkers of Choline Metabolism in Diffuse Adult Glioma: 18F-Fluoromethylcholine PET/CT, Magnetic Resonance Spectroscopy, and Choline Kinase α. Cancers. 2019;11:1969.

5. Why the authors selected Cho-PET, instead of [18F]FAZA PET, for example? I think that clarifying the advantages and disadvantages of this choice could increase the informative role of the present work.

For many years, the focus of our research group has been the role of various F-18 labeled PET-tracers in neuro-oncology. In comparison with other PET tracers, FDG and non-FDG tracers alike, [18F]FCho has certain advantages of which a very low uptake in normal white and grey matter of the brain is of major interest because it enhances the contrast between tumour and normal brain tissue. Secondly, it is – at least in Europe - widely available and it is still the tracer of choice in the management of castration resistant prostate cancer for those centers that do not have access to PSMA PET.

Although [18F]FCho PET has been studied for glioma imaging before by other groups, the number of studies is still limited compared to the number of publications on amino acid and hypoxia PET biomarkers.

Since hypoxia is associated with tumor aggressiveness, radiation resistance and poor prognosis, it is possible that changes in [18F]FAZA or [18F]FMISO uptake between pre- and post-treatment can be used to monitor treatment response. However, only few studies have been performed. One downside is that the degree of hypoxia can theoretically fluctuate, influenced by therapy and the presence of acute versus chronic hypoxia which can influence the reproducibility of hypoxia PET (Hirata et al. 2019, Mapelli P et al. 2017, Mönnich et al. 2012). In 2013, [18F]FAZA and [18F]FDG uptake in the F98 GB rat model was investigated by Belloli et al., however, not specifically for therapy response assessment. 

It is also noteworthy that there is no current consensus on which tracer is the best for hypoxia imaging. [18F]FAZA has advantages compared to [18F]FMISO due to better pharmacokinetic properties but [18F]FMISO can cross the blood–brain barrier because of its lipophilic nature while [18F]FAZA and [18F]DiFA can not (Hirata et al. 2019). Another downside of [18F]FAZA is that imaging is optimal 2-3 hours post-injection (with [18F]FMISO up to 4h), making it less convenient to work with. However, we studied [18F]FET and [18F]FAZA PET in another in vivo study by our group with a focus on the feasibility of PET-guided irradiation (Verhoeven et al. 2019). Because hypoxia is directly related to radiation resistance, [18F]FAZA PET could be used to guide an additional boost on hypoxic tumor regions as a strategy to overcome radioresistance and increase therapy effectiveness. [18F]FET has already been studied in depth for imaging glioma, however, tumor-to-normal brain contrast is less optimal compared to [18F]FCho.

The underlined parts have been included in the introduction (line 142-144). 

- Verhoeven J, Bolcaen J, De Meulenaere V, Kersemans K, Descamps B, Donche S, et al. Technical feasibility of [18F]FET and [18F]FAZA PET guided radiotherapy in a F98 glioblastoma rat model. Radiat Oncol. 2019;14(1):89.

- Mapelli P, Zerbetto F, Incerti E, Conte GM, Bettinardi V, Fallanca F, et al. 18F-FAZA PET/CT hypoxia imaging of high-grade glioma before and after radiotherapy. Clin Nucl Med 2017;42:e525-26.

- Belloli S, Brioschi A, Politi LS, Ronchetti F, Calderoni S, Raccagni I,et al. Characterization of biological features of a rat F98 GBM model: a PET-MRI study with [18F]FAZA and [18F]FDG. Nucl Med Biol. 2013;40:831-40.

- Hirata K, Yamaguchi S, Shiga T, Kuge Y, Tamaki N. The Roles of Hypoxia Imaging Using 18F-Fluoromisonidazole Positron Emission Tomography in Glioma Treatment. J Clin Med. 2019;8:1088. 

- Mönnich D, Troost EG, Kaanders JH, Oyen WJ, Alber M, Thorwarth D. Modelling and simulation of the influence of acute and chronic hypoxia on [18F]fluoromisonidazole PET imaging. Phys Med Biol. 2012;57:1675-84.

6. How where the simple size selected?statistical analysis were performed to selected a population of 10 animals?

Using statistical power analysis based on ANOVA (repeated measures, within-between interactions) using 2 group, 5 repeated measurements, an alpha-value of 0.05, a power of 0.80 and an effect size of 0.4, a total sample size of 10 animals was calculated.

7. Finally, the author disclosure a financial support by Lux Luka Foundation, but they must clearly state, according to Journal guidelines, who exactly received fundings, and the role of the sponsor in the study design and analysis.

Lux Luka Foundation did indeed support this study financially. Funds were received by Prof. I Goethals and Prof. T Boterberg. The sponsor did not play a role in study design and analysis. The funding source was not included in the Acknowledgements section in the manuscript, according to the Journal’s guidelines. 

However, this information was added in the cover letter.

---

## [Editor Report · Decision Letter 1]

22 Feb 2021

Assessment of the effect of therapy in a rat model of glioblastoma using [18F]FDG and [18F]FCho PET compared to contrast-enhanced MRI.

PONE-D-20-36019R1

Dear Dr. Bolcaen,

We’re pleased to inform you that your manuscript has been judged scientifically suitable for publication and will be formally accepted for publication once it meets all outstanding technical requirements.

Kind regards,

Pierpaolo Alongi

Academic Editor

PLOS ONE
---

## [Editor Report · Acceptance letter]

23 Feb 2021

PONE-D-20-36019R1 

Assessment of the effect of therapy in a rat model of glioblastoma using [^18^F]FDG and [^18^F]FCho PET compared to contrast-enhanced MRI. 

Dear Dr. Bolcaen:

I'm pleased to inform you that your manuscript has been deemed suitable for publication in PLOS ONE. Congratulations! Your manuscript is now with our production department. 

Kind regards, 

on behalf of

Dr. Pierpaolo Alongi 

Academic Editor

PLOS ONE